# Swiss Army Knife: Synergizing BiAses in Knowledge from Vision Foundation Models for Multi-Task Learning

**Yuxiang Lu**[1*]  **Shengcao Cao**[2*]  **Yu-Xiong Wang**[2]
[1]Shanghai Jiao Tong University  [2]University of Illinois Urbana-Champaign
`luyuxiang_2018@sjtu.edu.cn`  `{cao44,yxw}@illinois.edu`

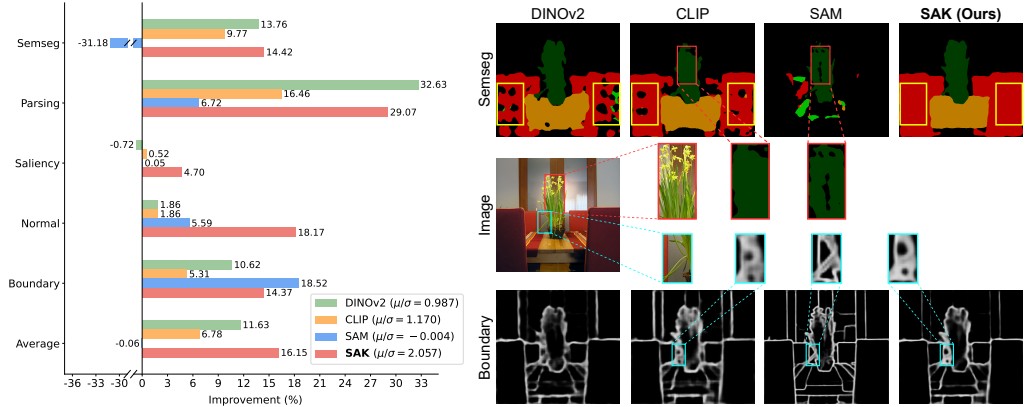

Figure 1: *(Left)* **Quantitative analysis of representation biases** in Vision Foundation Models (VFMs), including DINOv2, CLIP, and SAM, on the PASCAL-Context dataset across five vision tasks, all using the ViT-B backbones with pretrained parameters frozen. VFMs exhibit advantages and disadvantages across different downstream tasks when compared to a conventional ImageNet-pretrained backbone. Our SAK model, distilled from these VFM teachers, achieves the *best average performance with more balanced improvements*, as indicated by its larger ratio of mean improvement to standard deviation ($\mu/\sigma$). *(Right)* **Qualitative comparison of representation biases** through representative examples from semantic segmentation and boundary detection tasks. DINOv2 captures localized features but occasionally confuses semantic categories; CLIP excels in object-level understanding but lacks fine pixel-level details; SAM produces precise masks in both tasks due to higher input resolution but struggles with semantic knowledge. Our SAK successfully combines the *precise boundary detection* of SAM with the *accurate semantic understanding* of DINOv2 and CLIP. Further details are discussed in Section 2.

## Abstract

Vision Foundation Models (VFMs) have demonstrated outstanding performance on numerous downstream tasks. However, due to their inherent representation biases originating from different training paradigms, VFMs exhibit advantages and disadvantages across distinct vision tasks. Although amalgamating the strengths of multiple VFMs for downstream tasks is an intuitive strategy, effectively exploiting these biases remains a significant challenge. In this paper, we propose a novel and versatile "Swiss Army Knife" (SAK) solution, which adaptively distills knowledge from a committee of VFMs to enhance multi-task learning. Unlike existing methods that use a single backbone for knowledge transfer, our approach preserves the unique representation bias of each teacher by collaborating the lightweight Teacher-Specific Adapter Path modules with the Teacher-Agnostic Stem. Through dynamic selection and combination of representations with Mixture-of-Representations Routers, our SAK is capable of synergizing the complementary strengths of multiple VFMs. Extensive experiments show that our SAK remarkably outperforms prior state of the arts in multi-task learning by 10% on the NYUD-v2 benchmark, while also providing a flexible and robust framework that can readily accommodate more advanced model designs. Project page: https://innovator-zero.github.io/SAK/.

---

*Equal Contribution

# 1 INTRODUCTION

Vision Foundation Models (VFMs), such as DINOv2 (Oquab et al., 2024), CLIP (Radford et al., 2021), and SAM (Kirillov et al., 2023), have gained significant attention due to their impressive performance on various downstream tasks. This underscores the importance of integrating VFMs into Multi-Task Learning (MTL) (Caruana, 1997; Zhang & Yang, 2021; Yu et al., 2024), which aims to jointly learn multiple tasks with a single network, thereby enhancing model efficiency and performance, with broad applications in areas like autonomous driving (Ishihara et al., 2021).

In computer vision, multi-task models typically use a shared encoder to extract general features for all tasks, as they share a common interpretation of visual input (Ye & Xu, 2023a). A straightforward approach is to directly replace the encoder backbone with a VFM. However, VFMs are pretrained on diverse datasets, image resolutions, and objectives, introducing *representation biases* when applied as feature extractors for downstream tasks. Our empirical study in Figure 1 reveals that these inherent biases yield both advantages and disadvantages across different tasks, with *no single model achieving consistently superior performance across all domains*. These findings highlight the challenge of accomplishing comprehensive improvements in MTL using VFMs, pointing to the demand for collaborative utilization of multiple VFMs to exploit their complementary strengths.

Several existing works attempt an intuitive solution by extracting image features through multiple VFMs and then concatenating or fusing these features for later decoding (Lin et al., 2023; Kar et al., 2024; Zong et al., 2024; Tong et al., 2024a;b; Man et al., 2024). While this enhances visual encoding, it comes with a major drawback: The inference of all vision encoders drastically increases computational costs, along with the memory and storage requirements due to the large-scale parameters, rendering it less practical for real-world applications.

Therefore, recent works (Ranzinger et al., 2024b; Shang et al., 2024; Sariyildiz et al., 2024) propose more efficient frameworks by distilling multiple VFM teachers into a single student model, which can deliver competitive results on downstream benchmarks. Despite the progress, this many-to-one distillation risks eliminating the representation biases of the VFM teachers, potentially limiting the model's ability to capitalize on their individual strengths for specific tasks. Zong et al. (2024) further point out that biased information from VFMs can lead to performance degradation when naively fused. Moreover, when matching one student to multiple teachers, reconciling diverse biases in shared parameters could induce optimization conflicts. Our pilot study in Table 2 shows that the student trained by many-to-one distillation does not consistently surpass the teachers in their respective proficient tasks.

To overcome these limitations, we propose a novel approach named **SAK**, with the goal to build a **S**wiss **A**rmy **K**nife model from a committee of VFMs to synergize their complementary strengths and enhance performance across multiple downstream tasks. Considering the key challenge of *preserving the representation biases* while ensuring model efficiency for deployment, we introduce a multi-teacher knowledge distillation framework. This framework incorporates a shared Teacher-Agnostic Stem alongside multiple Teacher-Specific Adapter Path modules, which produce specialized representations aligned with each corresponding VFM teacher. During distillation, the Teacher-Agnostic Stem is optimized simultaneously by gradients from all VFM teachers, thereby capturing universal knowledge. Meanwhile, the Teacher-Specific Adapter Paths accommodate the heterogeneous representation biases of each teacher, explicitly learning their diverse model characteristics.

Building on the reproduction of representation biases, the next step is to amalgamate the committee's expertise by *exploiting the individual biases*. Specifically, we treat each group of representations as a knowledgeable expert and design a Mixture-of-Representations Router. This router dynamically weighs and combines the most relevant representations, bridging the gap between general-purpose knowledge and task-specific characteristics. The collaboration of these modules allows the student to harness both commonalities and differences of the teachers, facilitating smoother and more comprehensive knowledge transfer. Furthermore, SAK is a highly flexible framework that can further benefit from more advanced architectural designs (*e.g.*, stronger task-specific decoders) and more powerful models (*e.g.*, larger teachers), offering a general solution to multi-task visual learning.

Our contributions are summarized as follows:

Table 1: **Comparison of Vision Foundation Models.** Although all utilize the same Vision Transformer (ViT) backbone, they greatly differ in their training paradigms, including data, image resolutions, and training objectives, which lead to diverse representation biases.

| Model | Training Dataset | Dataset Size | Resolution | Objective |
|---|---|---|---|---|
| ViT (Dosovitskiy et al., 2021) | ImageNet-1k/21k | 1.2M/14.2M | 384 | Supervised classification |
| DINOv2 (Oquab et al., 2024) | LVD-142M | 142M | 518 | Discriminative self-supervised learning |
| CLIP (Radford et al., 2021) | WebImageText | 400M | 224 | Image-text contrastive learning |
| OpenCLIP (Cherti et al., 2023) | LAION-2B | 2B | 384 | Image-text contrastive learning |
| SAM (Kirillov et al., 2023) | SA-1B | 11M+1B | 1024 | Supervised promptable segmentation |

- We systematically analyze the distinct representation biases of Vision Foundation Models, which result in varying advantages and disadvantages across tasks, underscoring the importance of preserving these biases during distillation from multiple VFM teachers.
- We propose SAK, an efficient and effective solution that distills knowledge from VFM teachers into a Teacher-Agnostic Stem with Teacher-Specific Adapter Path modules, sharing common knowledge while retaining the biases. We also introduce Mixture-of-Representations Routers to adaptively amalgamate the complementary and specialized strengths for downstream tasks.
- We evaluate SAK on two widely-used multi-task benchmarks, PASCAL-Context and NYUD-v2, showing it remarkably outperforms previous multi-teacher VFM distillation methods and state-of-the-art multi-task models in both performance and robustness.
- SAK offers high flexibility and scalability, supporting a broad variety of VFM teachers and downstream tasks, and is compatible with various adapter, router, or decoder head architectures.

## 2 REPRESENTATION BIASES IN VISION FOUNDATION MODELS

In this section, we investigate the representation biases of Vision Foundation Models on multiple downstream tasks through empirical studies. We select three representative state-of-the-art VFMs: (1) *DINOv2* (Oquab et al., 2024), which claims to excel in dense prediction tasks such as semantic segmentation and depth estimation; (2) *CLIP* (Radford et al., 2021) and its reproduction, *OpenCLIP* (Cherti et al., 2023), which are recognized for capturing language-aligned semantics and employed as vision encoders in vision-language models; and (3) *SAM* (Kirillov et al., 2023), which achieves outstanding performance in promptable segmentation. For CLIP and SAM, we use only their image encoders for representation learning.

As summarized in Table 1, although all these VFMs utilize Vision Transformers (ViT) (Dosovitskiy et al., 2021) as backbones, they differ significantly in their training paradigms regarding datasets, dataset sizes, image resolutions, and training objectives. Consequently, *the representations learned by these models embed heterogeneous biases*, causing each model to focus on different aspects of image features and exhibit strengths and weaknesses in specific tasks.

We conduct comprehensive quantitative and qualitative experiments using the three VFMs on five dense prediction tasks from the PASCAL-Context dataset (Mottaghi et al., 2014). Among these tasks, intuitively, *semantic segmentation* and *human parsing* require high-level semantics of objects and localized features to generate accurate masks. *Saliency detection* demands an overall understanding of the image to identify its main contents, while *surface normal estimation* and *object boundary detection* depend more on fine-grained representations for precise predictions.

We further provide a pilot study to show the inferiority of ignoring representation biases in knowledge distillation from multiple VFM teachers, which validates the significance of addressing this problem and motivates the development of our methodology.

### 2.1 QUANTITATIVE ANALYSIS

To quantitatively analyze the representation biases, we evaluate the performance of the three VFMs directly transferred to each downstream task. We first freeze the models to generate image representations based on their pretrained knowledge, and then train a decoder head to produce final predictions for each task. DINOv2 and CLIP operate at a resolution of 512 on the downstream dataset, while SAM uses an input size of 1,024 as required by its pipeline. All feature maps are resized to 1/4 of the output resolution before being passed to the head. To quantify the advantages of VFMs over the conventional ImageNet-pretrained ViT backbone, we calculate their relative improvement over ViT for each task.

Table 2: **Comparison of a student model trained by many-to-one distillation without preserving representation biases and the oracle derived from VFM teachers.** The oracle selects the best result from the three teachers for each task. The student's 2.34% average underperformance demonstrates the critical importance of maintaining these biases during distillation.

| Model | Semseg mIoU↑ | Parsing mIoU↑ | Saliency maxF↑ | Normal mErr↓ | Boundary odsF↑ |
|---|---|---|---|---|---|
| Oracle of teachers | 81.18 (DINOv2) | 74.38 (DINOv2) | 81.48 (CLIP) | 16.21 (SAM) | 75.89 (SAM) |
| Student w/o biases | 80.18 (↓ 1.23%) | 69.13 (↓ 7.06%) | 82.72 (↑ 1.52%) | 16.00 (↑ 1.30%) | 71.16(↓ 6.23%) |

Figure 1*(Left)* illustrates how the representation biases in VFMs manifest in varying strengths and weaknesses in different downstream tasks. Specifically, DINOv2 shows significant improvements in two segmentation tasks, particularly excelling in human parsing with a performance gain of over 30%. It also performs well in object boundary detection, benefiting from its strongly localized features learned from the combination of image-level contrastive objective (Caron et al., 2021) and patch-level reconstructive objective iBOT (Zhou et al., 2022). While CLIP achieves lower accuracy than DINOv2 in these three tasks, it still exceeds the baseline by a notable margin of over 5%. Despite being pretrained on a segmentation task, SAM surprisingly underperforms ViT in semantic segmentation, showing a 30% drop, because of its limited semantic understanding—SAM considers solely the object masks and ignores their semantic labels in its promptable segmentation task. However, SAM is the best in surface normal estimation and object boundary detection, exhibiting strength in capturing pixel-level details and object edges. We also compute the ratio of mean improvement $\mu$ to standard deviation $\sigma$ across tasks, which can measure the consistency of improvements. A higher ratio indicates better outcomes, as it reflects larger average improvements with smaller dispersion. We can observe that while DINOv2 demonstrates stronger average enhancement, CLIP attains more balanced results, whereas SAM is inferior in both perspectives.

## 2.2 QUALITATIVE ANALYSIS

To validate our quantitative findings, we visualize the final predictions for semantic segmentation and boundary detection using an example image in Figure 1*(Right)*. We observe that DINOv2, while being effective at capturing localized features, is less effective than CLIP in semantic perception, as illustrated in the yellow box of Semseg results. In this case, DINOv2 confuses a chair with a sofa, resulting in misclassification as the background (the holes). Although CLIP excels in object-level understanding with its rich semantic knowledge from the language domain, it falls short in generating fine-grained pixel-level masks. This shortcoming arises because CLIP's training objective prioritizes image-level contents that are represented only by the class token, which possibly accounts for its lower performance than DINOv2 in the quantitative analysis.

On the other hand, SAM produces exceptional details in both tasks due to its high input resolution, as demonstrated in the red and cyan boxes. In the red box, a complex scene shows a foreground flower blending into the background, yet SAM accurately detects and labels the background in the segmentation mask. Notably, the background is not annotated in the ground truth, as such precise masking requires significant time and effort. We regard SAM's high resolution as its representation bias, as it stems directly from the model's training paradigm. However, SAM's limitation lies in its semantic knowledge, particularly when integrating semantics from multiple objects. This makes it difficult to attain high-quality semantic segmentation results, even with highly precise masks, echoing the quantitative analysis. We provide additional analysis and discussions in Appendix A.

## 2.3 IMPORTANCE OF PRESERVING REPRESENTATION BIASES

In summary, the inherent representation biases in VFMs result in their uneven performance across tasks, *with no single model achieving the best results in all areas*, as reflected in our quantitative analysis. This motivates the idea of combining multiple VFMs to achieve optimal performance in all tasks. Existing methods (Ranzinger et al., 2024b; Shang et al., 2024; Sariyildiz et al., 2024) propose a solution by distilling multiple VFMs into a single student model. However, given that the student model is shared by several teachers, an important question naturally arises: *Should we respect their individual representation biases when combining diverse VFMs?*

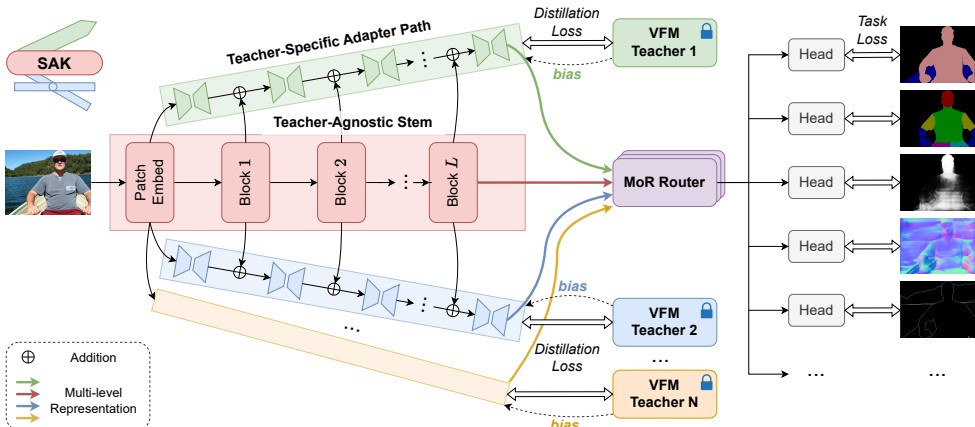

Figure 2: **Overview of our proposed SAK framework,** which distills foundational knowledge from a committee of frozen VFM teachers into an efficient student model. The student model operates like a Swiss Army Knife, with the Teacher-Agnostic Stem (TAS) serving as the main branch to learn universal knowledge among teachers. Each Teacher-Specific Adapter Path (TSAP) acts as a specialized tool to preserve the inherent representation bias of each teacher. Task-specific Mixture-of-Representations (MoR) Routers are then employed to synergize the complementary strengths of the teachers' biases, adaptively combining multi-level representations from both TAS and TSAP to generate tailored features for each task.

To answer this question, we conduct a pilot study where a student model is distilled from three aforementioned VFMs, using only linear aligners to match the student's features with those of the teachers, following the setup of the state-of-the-art method (Ranzinger et al., 2024b). In this approach, the representation biases are not explicitly preserved during distillation, leading the student to learn a unified representation aimed at simultaneously matching all three teachers. We then evaluate its performance on downstream tasks with the same settings as in quantitative analysis in Section 2.1. We compare the student without biases to an oracle derived from the teachers by selecting the best-performing teacher for each task, which represents the optimal performance of teachers.

From Table 2, the answer is clearly **YES**. While the distilled student surpasses the oracle in Saliency and Normal, somewhat validating the effectiveness of prior methods, it suffers from drastic performance degradation in Parsing and Boundary, with a drop of over 6%. Given that both the teachers and student utilize a ViT-B backbone in this study, the performance gap would likely widen with larger models. This demonstrates the limitation of naively transferring knowledge from multiple teachers into a student and highlights the importance of preserving the individual biases, leading us to the key question: ***Can we preserve the representation biases of multiple VFMs during distillation to maximize multi-task performance?*** Our methodology provides a positive answer to this challenge in the following sections.

## 3 METHODOLOGY

### 3.1 OVERVIEW

The overall framework of the proposed SAK is depicted in Figure 2. As a multi-teacher distillation approach, it employs a committee of VFM teachers, including DINOv2, CLIP, and SAM. The student model comprises a **Teacher-Agnostic Stem (TAS)** and multiple **Teacher-Specific Adapter Path (TSAP)** modules. TAS produces general representations shared across all branches, while each TSAP adapts the common representations to align with the specialized domain of its corresponding teacher via distillation. In this approach, the TSAP modules are optimized explicitly to replicate the unique representation biases of the teachers, all in a parameter- and computationally-efficient manner. The resulting feature sets are then passed through task-specific **Mixture-of-Representations (MoR) Routers** for adaptive combination and are finally processed by prediction heads to generate outputs for multiple tasks. We utilize multi-level representations for both the distillation and task decoding procedures, an essential aspect for dense prediction tasks (Ye & Xu, 2022).

## 3.2 TEACHER-AGNOSTIC STEM & TEACHER-SPECIFIC ADAPTER PATH

We adopt an off-the-shelf Vision Transformer (ViT) (Dosovitskiy et al., 2021) as the Teacher-Agnostic Stem (TAS) and design a lightweight network branch called Teacher-Specific Adapter Path (TSAP) parallel to the main stem. Given a TAS with $L$ blocks, the forward pass for an input image $\boldsymbol{X}$ is expressed as:

$$\boldsymbol{Z}_0 = \mathtt{PatchEmbed}(\boldsymbol{X}); \quad \boldsymbol{Z}_l = b_l(\boldsymbol{Z}_{l-1}), \quad l \in \{1, 2, \ldots, L\}, \tag{1}$$

where $b_l$ represents the $l$-th block, and $\boldsymbol{Z}_l \in \mathbb{R}^{n \times d}$ denotes its intermediate outputs with $n$ tokens of dimension $d$. Each TSAP module consists of $L+1$ adapters $\{a_l\}, l \in \{0, 1, \ldots, L\}$, with one adapter parallel to each patch embedding layer or transformer block. These adapters process intermediate features to adapt them to the teacher-specific representations $\boldsymbol{R}_l \in \mathbb{R}^{n \times d}$ in a residual manner:

$$\boldsymbol{R}_0 = a_0(\boldsymbol{Z}_0); \quad \boldsymbol{R}_l = a_l(\boldsymbol{R}_{l-1} + \boldsymbol{Z}_l), \quad l \in \{1, 2, \ldots, L\}. \tag{2}$$

We utilize the standard adapter structure (Houlsby et al., 2019), which includes a down-projection layer $\mathbf{W}_{\text{down}} \in \mathbb{R}^{d \times r}$, a GELU non-linearity (Hendrycks & Gimpel, 2016), and an up-projection layer $\mathbf{W}_{\text{up}} \in \mathbb{R}^{r \times d}$, where $r \ll d$ is the reduced dimension. As in prior works (Chen et al., 2022; Mercea et al., 2024), we integrate a learnable scaling factor $\alpha$ and a residual connection from the input $\boldsymbol{R}_{\text{in}} \in \mathbb{R}^{n \times d}$ to the output $\boldsymbol{R}_{\text{out}} \in \mathbb{R}^{n \times d}$, which can be formulated as:

$$\boldsymbol{R}_{\text{out}} = \alpha \mathtt{GELU}(\boldsymbol{R}_{\text{in}} \mathbf{W}_{\text{down}}) \mathbf{W}_{\text{up}} + \boldsymbol{R}_{\text{in}}. \tag{3}$$

For a committee of $N$ VFM teachers, we assign a TSAP module with adapters $\{a_l^i\}, l \in \{0, 1, \ldots, L\}$ to the $i$-th teacher. We then select four evenly distributed blocks from its outputs $\{\boldsymbol{R}_l^i\}$ to form multi-level representations $\{\boldsymbol{R}^i\}_s = \{\boldsymbol{R}_l^i\}, l \in \mathbb{L}_s = \{L/4, L/2, 3L/4, L\}$. Similarly, we have shared multi-level representations $\{\boldsymbol{Z}\}_s$ from TAS. Benefiting from the lightweight adapters, our distilled student model maintains original efficiency, as each TSAP module accounting for less than 5% of the TAS parameters. Consequently, our SAK framework is able to preserve the representation biases from the teachers without significant increases in computational cost, memory usage, or storage, as shown in detailed discussions in Appendix D.1.

## 3.3 MIXTURE-OF-REPRESENTATIONS ROUTER

As depicted in Figure 2, we treat the representations from TAS $\{\boldsymbol{Z}\}_s$ as a shared expert providing common knowledge, while the representations from each TSAP $\{\boldsymbol{R}^i\}_s, i \in \{1, 2, \ldots, N\}$ serve as proxy experts of VFMs with representation biases mirroring the teachers, resulting in a total of $N+1$ experts. To optimize the multi-task performance, we leverage the Mixture-of-Experts (MoE) mechanism (Jacobs et al., 1991), which adaptively produces task-specific features from this pool of general-purpose and specialized representations.

To facilitate this, task-specific router networks are trained to generate gate scores for each expert representation, which serve as the weights for a linear combination. As shown in Figure 6, the representations from different VFMs exhibit substantial variation in norm magnitudes. Thus, applying different weights for individual patches within an image can be less effective, as it may disturb the inherent patterns. To address this, we design a Mixture-of-Representations (MoR) Router, which differs from prior works by generating a global gating score across all patches.

Specifically, for each selected level $l \in \mathbb{L}_s$ and downstream task $t \in \mathbb{T}$, our MoR Router $r_l^t$ takes the teacher-agnostic representation $\boldsymbol{Z}_l \in \mathbb{R}^{n \times d}$ as input, projects its channel dimension to $N+1$ through a two-layer MLP, and then averages over $n$ patches to get a feature vector $\boldsymbol{h}_l^t \in \mathbb{R}^{N+1}$. To improve stability, we incorporate the noisy gating technique (Shazeer et al., 2017) by generating a noise vector $\boldsymbol{e}_l^t \in \mathbb{R}^{N+1}$ through an additional MLP. Then we compute the gating score $\boldsymbol{g}_l^t \in \mathbb{R}^{N+1}$:

$$\boldsymbol{g}_l^t = \mathtt{Softmax}(\boldsymbol{h}_l^t + \mathcal{N}(0, 1)\mathtt{Softplus}(\boldsymbol{e}_l^t)). \tag{4}$$

The output gating scores are used to calculate the weighted sum of representations at each output level. The fused features are then passed through task-specific heads for final predictions.

## 3.4 TRAINING PARADIGM

Our training paradigm contains two stages, with teacher parameters always frozen. In the first stage, we train the student model on the ImageNet-1k dataset (Deng et al., 2009; Russakovsky et al.,

2015), focusing on aligning the outputs of the TSAP modules with their respective VFM teachers. ImageNet is chosen due to its diverse and extensive image samples, providing a strong basis for effective knowledge transfer. To maintain fairness—given that conventional ViT backbones are also pretrained on ImageNet—we opt not to use other larger datasets like those utilized in VFMs and RADIO (Ranzinger et al., 2024b). Following previous findings (Ranzinger et al., 2024b; Shang et al., 2024), we employ a combination of cosine distance and smooth-L1 losses for distillation. Let $\boldsymbol{T}_l^i$ be the $i$-th teacher's representation at a selected level $l \in \mathbb{L}_s$, the overall distillation loss is:

$$\mathcal{L}_{\text{distill}}(\boldsymbol{X}) = \sum_{l \in \mathbb{L}_s} \sum_{i=1}^{N} \left( \alpha \mathcal{L}_{\cos}(\boldsymbol{R}_l^i, \boldsymbol{T}_l^i) + \beta \mathcal{L}_{\text{smooth-L1}}(\boldsymbol{R}_l^i, \boldsymbol{T}_l^i) \right). \tag{5}$$

where $\alpha = 0.9$ and $\beta = 0.1$ are weighting coefficients.

In the second stage, we continue training on the downstream multi-task datasets. The distillation loss is still included, allowing the VFM teachers to transfer more specialized knowledge related to the downstream data domain. This ensures that the representation biases are further secured in the student model; otherwise the biases could potentially be diminished due to the issue of catastrophic forgetting (French, 1999) during downstream fine-tuning. The overall loss is then formulated as:

$$\mathcal{L}(\boldsymbol{X}) = \gamma \mathcal{L}_{\text{distill}}(\boldsymbol{X}) + \sum_{t \in \mathbb{T}} w_t \mathcal{L}_t(\boldsymbol{X}, \boldsymbol{Y}_t), \tag{6}$$

where $\mathcal{L}_t(\boldsymbol{X}, \boldsymbol{Y}_t)$ is the task-specific loss for task $t$, computed using the ground truth $\boldsymbol{Y}_t$. The hyperparameter $\gamma$ balances the distillation loss and the task losses, with a default value of $1.0$ for simplicity, while $w_t$ adjusts the importance of each task. We set fixed $w_t$ values following the standard practice in MTL (Maninis et al., 2019; Kanakis et al., 2020).

## 4 EXPERIMENTS

### 4.1 EXPERIMENTAL SETUP

**Datasets.** We conduct experiments on two widely-used multi-task datasets: PASCAL-Context (Mottaghi et al., 2014) with five vision tasks and NYUD-v2 (Silberman et al., 2012) with four tasks. Details can be found in Appendix B.2.

**Implementation.** We employ a pretrained ViT backbone for TAS and use simple task-specific heads consisting of MLP and convolution layers for decoding. The VFM teachers are DINOv2, CLIP, and SAM with the ViT-L backbones, unless otherwise stated. More implementation details are provided in Appendix B to ensure reproducibility.

**Baselines.** To evaluate the effectiveness of our method, we consider three categories of baselines: (1) *Single-task baseline*, where individual models are trained for each task using the same ViT-initialized architecture, and *multi-task baseline*, where a shared encoder and task-specific heads are trained jointly. (2) *Multi-teacher VFM distillation approaches*, namely RADIO (Ranzinger et al., 2024b) and Theia (Shang et al., 2024). We use their released models as encoder backbones, coupled with the same task heads as ours. (3) *State-of-the-art MTL models*, which involves complicated encoder or decoder designs. We assess the overall performance of each model with MTL Gain $\Delta_m$ by calculating the average relative difference across all tasks compared to the single-task baseline (Maninis et al., 2019).

### 4.2 MAIN RESULTS

Figure 3 presents a comparison between our proposed SAK and representative baseline methods on both PASCAL-Context and NYUD-v2 datasets, with all methods using the ViT-B backbones. On PASCAL-Context, SAK greatly boosts the performance in Semseg and Parsing, achieving an overall improvement of 1.66% over the previous SOTA. On NYUD-v2, our method establishes a new milestone across all four tasks, increasing the MTL Gain metric from the previous best of 6.33% to 11.11%. We provide more comprehensive comparisons on both datasets using the ViT-L backbones in Tables 5 and 6, and ViT-S/Swin-S backbones in Appendix C. Our approach consistently outperforms previous methods, achieving the best results on 7 out of 9 tasks and both MTL Gain metrics. Notably, SAK significantly surpasses the SOTAs in MTL (BFCI (Zhang et al., 2023b), MLoRE (Yang et al., 2024d)) by nearly 10% on NYUD-v2, all while using fewer parameters.

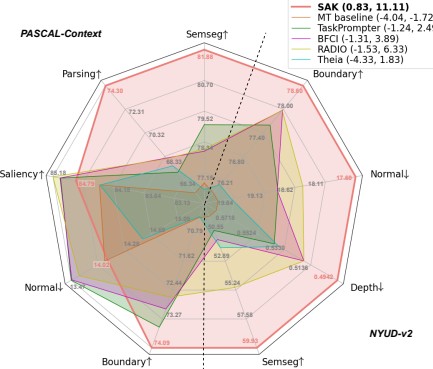

Figure 3: **Performance comparison on two datasets,** based on ViT-B backbones. MTL Gain $\Delta_m$ on two datasets are shown in the legend, respectively.

Table 3: **Ablation of proposed modules.** '↑': higher is better; '↓': lower is better; '$\Delta_m$': MTL Gain w.r.t. single-task baseline. 'Rep Sim' denotes the average cosine similarity between the representations of student and corresponding teachers on the ImageNet-1k validation set.

| TSAP | MoR | Rep Sim↑ | Semseg mIoU↑ | Parsing mIoU↑ | Saliency maxF↑ | Normal mErr↓ | Boundary odsF↑ | $\Delta_m\%$↑ |
|------|-----|----------|--------------|---------------|----------------|--------------|----------------|---------------|
| ✗ | ✗ | 0.3344 | 80.97 | 69.71 | 84.64 | 14.11 | 72.82 | -1.21 |
| ✔ | ✗ | 0.8708 | 81.26 | 69.92 | 84.31 | 14.45 | 71.41 | -2.03 |
| ✔ | ✔ | 0.8708 | **81.65** | **72.38** | **84.87** | 14.05 | **73.23** | **-0.03** |

Table 4: **Ablation of our two-stage training paradigm.** 'Distill': distillation loss; 'Task': task-specific losses.

| Stage1 | Stage2 Distill | Stage2 Task | Semseg mIoU↑ | Parsing mIoU↑ | Saliency maxF↑ | Normal mErr↓ | Boundary odsF↑ | $\Delta_m\%$↑ |
|--------|--------|------|--------------|---------------|----------------|--------------|----------------|---------------|
| ✗ | ✗ | ✔ | 76.76 | 65.26 | 84.39 | 13.98 | 70.37 | -4.04 |
| ✗ | ✔ | ✔ | 77.06 | 65.08 | 84.67 | **13.83** | 70.74 | -3.63 |
| ✔ | ✗ | ✔ | 80.48 | 71.16 | **85.04** | 13.92 | 72.60 | -0.60 |
| ✔ | ✔ | ✔ | **81.65** | **72.38** | 84.87 | 14.05 | **73.23** | **-0.03** |

Table 5: **Comparison with state of the arts on PASCAL-Context,** based on ViT-L backbones.

| Model | Backbone | #Param | Semseg mIoU↑ | Parsing mIoU↑ | Saliency maxF↑ | Normal mErr↓ | Boundary odsF↑ | $\Delta_m\%$↑ |
|-------|----------|--------|--------------|---------------|----------------|--------------|----------------|---------------|
| Single-task baseline | ViT-L | 1573M | 81.61 | 72.77 | 83.80 | 13.87 | 75.24 | 0.00 |
| Multi-task baseline | ViT-L | 357M | 79.26 | 68.28 | 84.16 | 14.06 | 71.59 | -2.97 |
| PAD-Net (Xu et al., 2018) | ViT-L | 330M | 78.01 | 67.12 | 79.21 | 14.37 | 72.60 | -4.95 |
| MTI-Net (Vandenhende et al., 2020) | ViT-L | 851M | 78.31 | 67.40 | 84.75 | 14.67 | 73.00 | -3.81 |
| ATRC (Brüggemann et al., 2021) | ViT-L | 340M | 77.11 | 66.84 | 81.20 | 14.23 | 72.10 | -4.71 |
| InvPT (Ye & Xu, 2022) | ViT-L | 423M | 79.03 | 67.61 | 84.81 | 14.15 | 73.00 | -2.81 |
| InvPT++ (Ye & Xu, 2024) | ViT-L | 421M | 80.22 | 69.12 | 84.74 | 13.73 | 74.20 | -1.19 |
| TaskPrompter (Ye & Xu, 2023b) | ViT-L | 401M | 80.89 | 68.89 | 84.83 | 13.72 | 73.50 | -1.24 |
| TaskExpert (Ye & Xu, 2023a) | ViT-L | 420M | 80.64 | 69.42 | 84.87 | 13.56 | 73.30 | -0.97 |
| BFCI (Zhang et al., 2023b) | ViT-L | 477M | 80.64 | 70.06 | 84.64 | 13.82 | 72.96 | -1.32 |
| 3D-aware (Li et al., 2024a) | ViT-L | 430M | 79.53 | 69.12 | 84.94 | 13.53 | 74.00 | -1.08 |
| TSP (Wang et al., 2024b) | ViT-L | 423M | 81.48 | 70.64 | 84.86 | 13.69 | 74.80 | -0.22 |
| MLoRE (Yang et al., 2024d) | ViT-L | 407M | 81.41 | 70.52 | 84.90 | 13.51 | 75.42 | 0.16 |
| RADIO (Ranzinger et al., 2024b) | ViT-L | 372M | 81.11 | 71.50 | **85.17** | **13.49** | 74.80 | 0.29 |
| SAK (Ours) | ViT-L | 407M | **84.01** | **76.99** | 84.65 | 13.82 | **76.27** | **2.30** |

## 4.3 IN-DEPTH ANALYSIS

We conduct extensive experiments to validate the effectiveness and generalization of our proposed SAK framework. All experimental analyses are based on the ViT-B backbones for both teachers and student and the PASCAL-Context dataset unless otherwise specified.

**Ablation study.** An ablation study is conducted to discern the individual contributions of the main components in SAK, namely TSAP and MoR Router, as outlined in Table 3. We consider two model variants: (1) a model without the TSAP and MoR Router modules (row 1), which corresponds to the student distilled naively regardless of representation biases, as studied in Table 2; (2) a model distilled with TSAP in the first stage but trained without MoR Routers in the second stage (row 2), where the biased representations from multiple VFMs are simply added together. Firstly, our results confirm that our proposed TSAP effectively preserves the representation biases from the teachers as indicated by a higher average similarity between the student and teachers. Additionally, we prove that a simple fusion of diverse biased knowledge does not lead to an overall improvement and may even fall behind compared to the student without biases. With the synergization of TSAP and MoR Router, our proposed SAK not only preserves and reproduces the representation biases after distillation but also optimally capitalizes on these biases to maximize multi-task performance. The upper-bound results of teacher amalgamation are presented in Appendix C.

Table 4 reports another ablation on our training paradigm, highlighting the contributions of each stage. The results show that Stage 1, which distills knowledge from VFM teachers on ImageNet, is a primary factor of performance enhancement. Meanwhile, incorporating the distillation loss during Stage 2 consistently boosts final outcomes, regardless of whether Stage 1 is applied. This underscores the effectiveness of transferring specialized knowledge in the downstream data domain.

Table 6: **Comparison with state of the arts on NYUD-v2,** based on ViT-L backbones.

| Model | Backbone | #Param | Semseg mIoU↑ | Depth RMSE↓ | Normal mErr↓ | Boundary odsF↑ | $\Delta_m\%$ ↑ |
|---|---|---|---|---|---|---|---|
| Single-task baseline | ViT-L | 1259M | 54.19 | 0.5560 | 19.22 | 78.09 | 0.00 |
| Multi-task baseline | ViT-L | 346M | 52.42 | 0.5413 | 19.29 | 76.50 | -0.76 |
| InvPT (Ye & Xu, 2022) | ViT-L | 402M | 53.56 | 0.5183 | 19.04 | 78.10 | 1.64 |
| InvPT++ (Ye & Xu, 2024) | ViT-L | ~402M | 53.85 | 0.5096 | 18.67 | 78.10 | 2.65 |
| TaskPrompter (Ye & Xu, 2023b) | ViT-L | 392M | 55.30 | 0.5152 | 18.47 | 78.20 | 3.36 |
| TaskExpert (Ye & Xu, 2023a) | ViT-L | 400M+ | 55.35 | 0.5157 | 18.54 | 78.40 | 3.33 |
| BFCI (Zhang et al., 2023b) | ViT-L | 400M+ | 55.51 | 0.4930 | 18.47 | 78.22 | 4.46 |
| 3D-aware (Li et al., 2024a) | ViT-L | 409M | 54.87 | 0.5006 | 18.55 | 78.30 | 3.74 |
| TSP (Wang et al., 2024b) | ViT-L | 402M | 55.39 | 0.4961 | 18.44 | 77.50 | 4.07 |
| MLoRE (Yang et al., 2024d) | ViT-L | 552M | 55.96 | 0.5076 | 18.33 | 78.43 | 4.26 |
| RADIO (Ranzinger et al., 2024b) | ViT-L | 362M | 59.32 | 0.4698 | 17.46 | 79.41 | 8.95 |
| SAK (Ours) | ViT-L | 394M | **63.18** | **0.4313** | **16.25** | **79.43** | **14.05** |

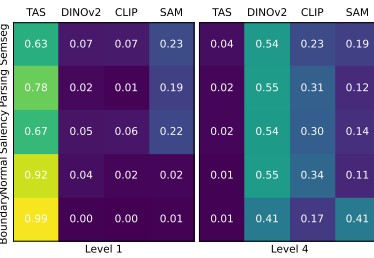

Figure 4: **Weights of different experts learned by MoR Routers.**

Table 7: **Performance w.r.t. different combinations of VFM teachers.** Integrating knowledge from three teachers leads to the strongest overall performance.

| Teachers | Semseg mIoU↑ | Parsing mIoU↑ | Saliency maxF↑ | Normal mErr↓ | Boundary odsF↑ | $\Delta_m\%$ ↑ |
|---|---|---|---|---|---|---|
| Multi-task baseline | 76.76 | 65.26 | 84.39 | 13.98 | 70.37 | -4.04 |
| DINOv2 | 79.05 | 69.55 | 84.29 | 14.07 | 71.14 | -2.21 |
| CLIP | 80.12 | 67.57 | 83.81 | 14.41 | 70.62 | -3.26 |
| SAM | 63.47 | 63.99 | **85.02** | **13.95** | **73.27** | -6.74 |
| DINOv2+CLIP | 81.53 | 71.95 | 84.49 | 14.16 | 72.59 | -0.60 |
| DINOv2+CLIP+SAM | **81.65** | **72.38** | 84.87 | 14.05 | 73.23 | **-0.03** |

**Impact of teacher selection.** To investigate whether knowledge from all teachers can be effectively incorporated into the student model and how each teacher contributes to downstream tasks, we experiment on different combinations of VFM teachers in Table 7. When using a single teacher, SAK effectively learns the teacher's representation bias, as the student distilled from DINOv2 or CLIP performs well in segmentation tasks, while SAM's student is better in tasks requiring finer details. Combining DINOv2 and CLIP continues to improve segmentation tasks, potentially due to their complementary strengths in localized feature learning and semantic understanding. Including SAM further benefits all tasks, leading to the best overall results. We also visualize the gating weights learned by our proposed MoR Routers at the lowest and highest levels in Figure 4. At the lowest level, where VFM teachers share more general knowledge about the details, tasks tend to rely on the shared TAS and SAM's bias. Conversely, the representation biases become more pronounced at higher levels; therefore, the teacher-specific representations are predominantly selected. Further analysis is provided in Appendix D.3 and D.4.

**Impact of downstream data size.** To assess the robustness of multi-teacher VFM distillation methods, we conduct experiments using varying numbers of samples among {25%, 50%, 75%, 100%} from the downstream dataset. As depicted in Figure 5, while all models show an upward trend as the number of data samples increases, our SAK consistently outperforms the other two distillation baselines across all settings. Particularly, SAK surpasses the second-best method by a clear margin of over 3% in scenarios with substantially fewer samples such as merely 25%.

**Scaling with model size.** In Table 8, we explore the impact of scaling the backbone sizes of the VFM teachers and student by forming various combinations. The results indicate that increasing the capacity of the student model, while keeping the teacher models fixed (row 1 *vs.* row 2, row 3 *vs.* row 4), yields remarkable improvements across nearly all tasks. Additionally, scaling up the teacher models without altering the student (row 2 *vs.* 3) also proves beneficial. These results demonstrate the versatility and robustness of our approach in adapting to models of varying sizes.

**Compatibility with different decoders.** It is worth noting that our SAK framework is flexible and does not impose constraints on the design of the backbone, the adapters in TSAP, or the decoder heads. As shown in Table 9, we replace the simple head with the more complex MLoRE decoder (Yang et al., 2024d). Even with a simple head, SAK surpasses MLoRE by 0.8%, and integrating the MLoRE decoder further enhances overall performance by 1.72%.

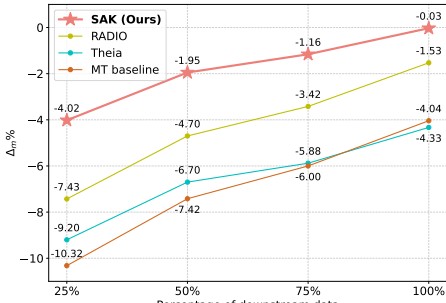

**Figure 5: Performance w.r.t. downstream data percentage.** MTL Gain is computed w.r.t. single-task baseline on full dataset. SAK is the most robust in downstream tasks.

Table 8: **Performance w.r.t. different settings of teacher and student sizes.** SAK shows robustness across teachers or students with varying capacities.

| Backbone | | Semseg | Parsing | Saliency | Normal | Boundary |
|---|---|---|---|---|---|---|
| Teachers | Student | mIoU↑ | mIoU↑ | maxF↑ | mErr↓ | odsF↑ |
| ViT-B | ViT-S | 78.66 | 68.46 | 84.66 | 14.33 | 70.28 |
| ViT-B | ViT-B | 81.65 | 72.38 | 84.87 | 14.05 | 73.23 |
| ViT-L | ViT-B | 81.88 | 74.30 | 84.79 | 14.02 | 74.09 |
| ViT-L | ViT-L | 84.01 | 76.99 | 84.65 | 13.82 | 76.27 |

Table 9: **Performance of SAK integrated with MLoRE.** SAK further benefits from stronger decoders.

| Enc. | Dec. | Semseg mIoU↑ | Parsing mIoU↑ | Saliency maxF↑ | Normal mErr↓ | Boundary odsF↑ | $\Delta_m$%↑ |
|---|---|---|---|---|---|---|---|
| ViT | MLoRE | 79.26 | 67.82 | **85.31** | **13.65** | 74.69 | -0.83 |
| SAK | Simple | 81.65 | 72.38 | 84.87 | 14.05 | 73.23 | -0.03 |
| SAK | MLoRE | **82.74** | **74.28** | 84.58 | 13.89 | **75.96** | **1.69** |

## 5 RELATED WORK

**Knowledge Distillation of Vision Foundation Models.** As large-scale generalists, Vision Foundation Models (VFMs) show superior performance in various tasks with minimal tuning, such as CLIP (Radford et al., 2021) for vision-language tasks, DINOv2 (Oquab et al., 2024) for fine-grained recognition, and SAM (Kirillov et al., 2023) for promptable segmentation. To reduce their computational demands while preserving performance, knowledge distillation (Buciluă et al., 2006; Hinton et al., 2014) has been widely adopted in compressing VFMs (Vemulapalli et al., 2024; Sun et al., 2023; Yang et al., 2024a). More recently, multiple VFMs are distilled into a single student to combine their strengths: SAM-CLIP (Wang et al., 2024a) merges CLIP into SAM via continual learning and distillation. RADIO (Ranzinger et al., 2024b) learns from CLIP, DINOv2, and SAM to enhance performance on downstream tasks. Theia (Shang et al., 2024) further incorporates Depth Anything (Yang et al., 2024b), showing advantages in robot learning. Different from the straightforward distillation in these methods, we adaptively transfer knowledge from multiple teachers while retaining the unique representation biases to maximize their strengths for multiple tasks.

**Multi-Task Learning.** Multi-Task Learning (MTL) aims to train a single model capable of handling multiple tasks simultaneously (Caruana, 1997; Zhang & Yang, 2021; Yu et al., 2024). MTL research primarily falls into two categories: multi-task optimization (Kendall et al., 2018; Chen et al., 2018; Yu et al., 2020) and model architecture design (Long et al., 2017; Wallingford et al., 2022; Lu et al., 2024c). Considering vision tasks, most works center on designing architectures, which is further divided into encoder-focused and decoder-focused methods (Vandenhende et al., 2021). Encoder-focused methods develop encoders to extract features for different tasks (Misra et al., 2016; Ruder et al., 2019; Gao et al., 2019), while decoder-focused methods introduce task-interaction modules in decoder to better capture task-specific features (Ye & Xu, 2022; Xu et al., 2023c; Ye & Xu, 2023b).

Knowledge distillation has also been applied to enhance MTL (Li & Bilen, 2020; Jacob et al., 2023; Ghiasi et al., 2021; Luo et al., 2020; Ye et al., 2019a). These methods train a multi-task model to mimic multiple single-task teachers, allowing the student to gain richer information. Xu et al. (2023d) propose directly distilling a small multi-task student from a large multi-task teacher. To the best of our knowledge, our work is the first exploration of multi-task distillation with general-purpose knowledge from task-agnostic VFM teachers, as opposed to task-related teachers trained on target datasets.

## 6 CONCLUSION

Building on our analysis of the representation biases in VFMs, we introduce a novel framework SAK, designed to improve multi-task learning by exploiting the complementary biases of multiple VFMs. Through the integration of a Teacher-Agnostic Stem, Teacher-Specific Adapter Paths, and Mixture-of-Representations Routers, SAK effectively preserves the unique representation biases during distillation, thereby enhancing both accuracy and robustness across multiple downstream tasks. Our work opens possibilities for including more advanced teachers and students, and provides a solid foundation for future advancements in multi-task visual learning with foundation models.

ACKNOWLEDGMENTS

This work was supported in part by NSF Grant 2106825, NIFA Award 2020-67021-32799, the Toyota Research Institute, the IBM-Illinois Discovery Accelerator Institute, the Amazon-Illinois Center on AI for Interactive Conversational Experiences, Snap Inc., and the Jump ARCHES endowment through the Health Care Engineering Systems Center at Illinois and the OSF Foundation. This work used computational resources, including the NCSA Delta and DeltaAI supercomputers through allocations CIS230012 and CIS240428 from the Advanced Cyberinfrastructure Coordination Ecosystem: Services & Support (ACCESS) program, as well as the TACC Frontera supercomputer and Amazon Web Services (AWS) through the National Artificial Intelligence Research Resource (NAIRR) Pilot.

ETHICS STATEMENT

Our research adheres to high ethical standards in machine learning and computer vision, ensuring transparency, reproducibility, and fairness in all experiments. While our approach shows promising results, it shares common limitations with other foundation models, particularly regarding data usage. We utilize publicly available datasets in compliance with relevant legal and ethical guidelines, emphasizing proper data handling during selection and pre-processing.

REPRODUCIBILITY STATEMENT

We ensure reproducibility by elaborating implementation details, which includes details on models, datasets, and training setups in Appendix B. Our code and models are publicly available at https://github.com/innovator-zero/SAK.

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

The appendix is organized as follows:

## A  ADDITIONAL ANALYSIS OF REPRESENTATION BIASES

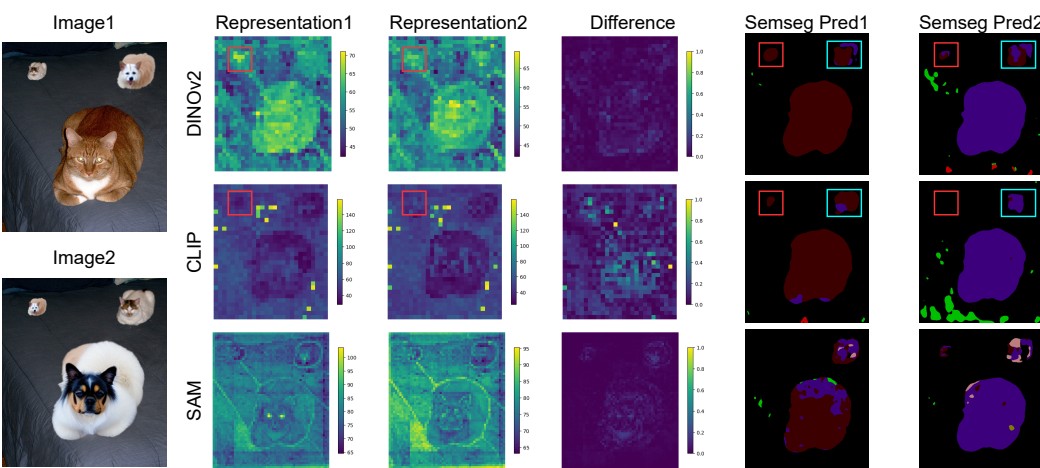

Figure 6: **Qualitative analysis of representation biases in three VFMs based on a pair of manipulated images.** We visualize the representations by calculating the L2 norm, their differences in response to content changes, and their semantic segmentation predictions. CLIP struggles to detect small objects, as they resemble background tokens, leading to poor segmentation; in contrast, DINOv2 excels in small object detection but confuses semantic features in complex scenarios, while SAM emphasizes edges, enhancing pixel-level detail but limiting high-level understanding.

Based on our analysis in Section 2, we further delve into the representations generated by VFMs to understand how the representation biases lead to different outcomes across tasks. In Figure 6, we create a pair of manipulated images by selecting several cat images from the dataset and converting them into dogs while preserving the same shape and pose using ControlNet (Zhang et al., 2023c). We place various sizes of cats and dogs in a shared background image to compose complex and challenging scenes. For the segmentation task, this ensures that the predictions should differ only in labels, while the mask shapes remain the same. We visualize the representations by calculating the L2 norm of each patch token and their representation differences.

The red boxes highlight why CLIP struggles to detect small objects, such as the smallest cat and dog. In the representations, small objects exhibit a pattern similar to neighboring background tokens, indicating that CLIP assigns them the same level of importance as the background. Consequently, CLIP

fails to segment these small objects and also struggles to generate precise masks for medium-sized objects, as shown in the cyan boxes. Conversely, DINOv2 performs better at detecting small objects by leveraging representations where these objects receive more attention than the background. However, its semantic understanding is limited in complex scenarios. For instance, DINOv2 confuses the heads of the cat and dog in the cyan boxes, even though this is the most distinguishable feature between the two animals. SAM displays different behaviors, with its representations concentrating more on the edges in the images instead of the entire objects, which explains its strength in capturing pixel-level details and weakness in high-level knowledge, as mentioned in our quantitative analysis. When examining the differences between the representations of the two images, CLIP appears more sensitive to changes in the main objects than DINOv2 and SAM, further underscoring its semantic focus. We also observe the phenomenon of register tokens (Darcet et al., 2024) in CLIP's representation, where these artifacts correspond to areas the model deems low-informative background.

Additionally, we extend our findings by providing insightful explanations that link the underlying causes (differences in training paradigms) and effects (variations in downstream performance) of the representation biases:

- *DINOv2* excels at capturing features across multiple scales, stemming from its combination of image-level contrastive objective (Caron et al., 2021) and patch-level reconstructive objective iBOT (Zhou et al., 2022). Although it lacks semantic-based supervision, its general-purposed visual representations exhibit powerful generalization capability for most downstream tasks.
- *CLIP* is trained by aligning the image and text embeddings of numerous image-text pairs through contrastive learning, which incorporates rich semantic knowledge from language domain that enhances image-level understanding. However, CLIP tends to prioritize prominent objects in an image while overlooking smaller ones, because (1) the alignment only considers the global class token (patch tokens are optimized implicitly), and (2) the image captions mainly focus on primary contents. This possibly accounts for its lower performance compared to DINOv2 in our quantitative experiments.
- *SAM*'s representation bias can be explained by two reasons. First, SAM is supervised by the promptable segmentation task, considering solely the object masks and ignoring their semantic labels. Second, SAM operates with an input resolution of 1,024 in both training and inference, producing nearly four times more patches than other models. Consequently, SAM is adept at capturing pixel-level details and detecting object edges, but is poor at perceiving the semantics of multiple objects.

## B  IMPLEMENTATION DETAILS

### B.1  MODELS

**Architecture.** We use DINOv2 (Oquab et al., 2024), CLIP (Radford et al., 2021), and SAM (Kirillov et al., 2023) as our VFM teachers, with pretrained models from timm (Wightman, 2019). For CLIP, we utilize the OpenCLIP (Cherti et al., 2023) reproduction in our experiments. The model names of the VFMs and multi-teacher VFM distillation baselines, RADIO (Ranzinger et al., 2024b)[1] and Theia (Shang et al., 2024)[2], are listed in Table 10. For the Teacher-Agnostic Stem in the SAK student model, we employ ViT-S, ViT-B, and ViT-L backbones, initialized with pretrained weights from timm. While we keep the `CLS` tokens in the model, we only use patch tokens as representations for both the teachers and student models. Other modules in the student are trained from scratch. The adapters in the Teacher-Specific Adapter Path use a channel reduction ratio of 4.

**Input resolution.** As detailed in Table 1, the VFMs are pretrained on diverse image resolutions. Additionally, RADIO-B is pretrained on $768 \times 768$, RADIO-L is pretrained on $1024 \times 1024$, and Theia on $224 \times 224$. In our experiments, we use a unified input size for all models, except SAM. In the first stage, we use an input size of $384 \times 384$, while in the second stage, the input size is set to $512 \times 512$ for PASCAL-Context and $448 \times 576$ for NYUD-v2. For SAM, we pad and resize all images to $1024 \times 1024$. Feature maps from all models are interpolated to $1/4$ of the output resolution before being decoded by the heads for a fair comparison.

---

[1] https://github.com/NVlabs/RADIO
[2] https://huggingface.co/collections/theaiinstitute/theia-66a7a6ae80a707547c358cce

Table 10: Model zoo of VFMs and multi-teacher VFM distillation methods.

| Model | Platform | Backbone | Name |
|---|---|---|---|
| ViT (Dosovitskiy et al., 2021) | timm | ViT-S/16 | `vit_small_patch16_384` |
| | | ViT-B/16 | `vit_base_patch16_384` |
| | | ViT-L/16 | `vit_large_patch16_384` |
| DINOv2 (Oquab et al., 2024) | timm | ViT-B/14 | `vit_base_patch14_dinov2` |
| | | ViT-L/14 | `vit_large_patch14_dinov2` |
| CLIP (Radford et al., 2021) OpenCLIP (Cherti et al., 2023) | timm | ViT-B/16 | `vit_base_patch16_clip_384` |
| | | ViT-L/14 | `vit_large_patch14_clip_336` |
| SAM (Kirillov et al., 2023) | timm | ViT-B/16 | `samvit_base_patch16` |
| | | ViT-L/16 | `samvit_large_patch16` |
| RADIO (Ranzinger et al., 2024b) | torch hub | ViT-B/16 | `NVlabs/RADIO/radio_v2.5-b` |
| | | ViT-L/16 | `NVlabs/RADIO/radio_v2.5-l` |
| Theia (Shang et al., 2024) | huggingface | ViT-S/16 | `theaiinstitute/theia-small-patch16-224-cdiv` |
| | | ViT-B/16 | `theaiinstitute/theia-base-patch16-224-cdiv` |

**Teacher/student mismatch.** Since we have VFM teachers operating with different input and patch sizes (14 or 16), their intermediate representations have different spatial resolutions. Take an input size of $512 \times 512$ as an example, DINOv2-B/14 teacher outputs a $36 \times 36$ map, CLIP-B/16 outputs a $32 \times 32$ map, SAM-B/16 outputs a $64 \times 64$ map, while the student based on ViT-B/16 outputs a $32 \times 32$ map. Moreover, the number of channels may differs between teachers and students when using distinct backbones, such as 1024 for ViT-L teacher and 768 for ViT-B student. To align the student's representations with the teachers', we introduce an upsampling layer to match the spatial resolution, and a linear layer for channel mapping when necessary. Note that the linear layer is only required during distillation, adding no overhead to the inference.

## B.2 DATASETS

In the first stage, we perform knowledge distillation using the ImageNet-1k dataset (Deng et al., 2009; Russakovsky et al., 2015), which contains 1.2 million images. Only the images are used, without any category labels. To evaluate multi-task learning performance, we use two benchmark datasets in the second stage: PASCAL-Context (Mottaghi et al., 2014) and NYUD-v2 (Silberman et al., 2012). PASCAL-Context contains 4,998 training samples and 5,105 testing samples, annotated for five tasks of semantic segmentation ('Semseg'), human parsing ('Parsing'), saliency detection ('Saliency'), surface normal estimation ('Normal'), and object boundary detection ('Boundary'). Among them, the surface normal and saliency labels are supplemented by previous work (Maninis et al., 2019). Meanwhile, NYUD-v2 consists of 795 images for training and 654 images for testing, all from indoor scenes. It provides labels for four tasks: semantic segmentation, monocular depth estimation ('Depth'), surface normal estimation, and object boundary detection.

## B.3 DATA AUGMENTATION

In the first stage, we apply standard image augmentation techniques, including random resizing and cropping to $384 \times 384$, along with random horizontal flipping. For the downstream datasets, we follow the established data augmentation protocols from previous studies (Maninis et al., 2019; Kanakis et al., 2020; Ye & Xu, 2022). We use random scaling with factor between 0.5 and 2.0, random cropping to the required input resolution ($512 \times 512$ for PASCAL-Context and $448 \times 576$ for NYUD-v2), random horizontal flipping, and random color jittering. Image normalization is also applied throughout both the training and evaluation phases.

## B.4 EVALUATION METRICS

We adopt evaluation metrics in previous methods (Ye & Xu, 2022). Specifically, we measure semantic segmentation and human parsing using the mean Intersection over Union (mIoU). Saliency detection is evaluated by the maximum F-measure (maxF). Surface normal estimation is evaluated by the mean error (mErr) of angles. Object boundary detection is evaluated by the optimal-dataset-scale F-measure (odsF) (Martin et al., 2004), implemented with the SEISM package (Pont-Tuset & Marques, 2015). The maximum allowed mis-localization for odsF is set to 0.0075 for PASCAL-

Context and 0.011 for NYUD-v2. Monocular depth estimation is evaluated by the Root Mean Square Error (RMSE).

To assess overall performance, we calculate the MTL Gain by determining the average relative difference of multi-task method compared to the single-task baseline (Maninis et al., 2019). The formula is as follows:

$$\Delta_m = \frac{1}{T} \sum_{t=1}^{T} (-1)^{l_t} \frac{M_t - M_{\mathrm{ST},t}}{M_{\mathrm{ST},t}}, \tag{7}$$

where $T$ is the number of tasks, $M_t$ and $M_{\mathrm{ST},t}$ represent the performance of task $t$ for the multi-task method and the single-task baseline, respectively. The value of $l_t = 1$ if a lower value means better performance for task $t$, and $l_t = 0$ otherwise.

### B.5 LOSS FUNCTIONS AND WEIGHTS

Following previous practices in the field (Maninis et al., 2019; Kanakis et al., 2020; Ye & Xu, 2022), our method employs task-specific loss functions. For semantic segmentation and human parsing, we use the cross-entropy loss, while for saliency detection, we apply the balanced cross-entropy loss. Surface normal estimation and monocular depth estimation are optimized using the L1 loss. For edge detection, we use the weighted binary cross-entropy loss, with a weight of 0.95 to positive pixels and 0.05 to negative ones. To balance multiple task losses, we compute a weighted sum of the task-specific losses. The loss weights for Semseg, Parsing, Saliency, Normal, Boundary, and Depth tasks are set to 1, 2, 5, 10, 50, and 1, respectively.

### B.6 TRAINING

As outlined in Section 3.4, the first stage trains the student model, which consists of the TAS and TSAP modules, with the distillation loss. Then in the second stage, we add the MoR Routers and prediction heads, training the entire student model with a combination of distillation loss and task-specific losses. The loss balancing factor $\gamma$ in Equation 6 is set as 1.0 for simplicity. Note that VFM teachers are always frozen in both stages. The detailed hyperparameter configurations are listed in Table 11. For the first stage, we follow the setup of RADIO (Ranzinger et al., 2024b) for optimizer, learning rate, LR scheduler, and distillation loss in Equation 5, while adopting Theia's (Shang et al., 2024) batch size configuration since we use the same dataset. To enhance training efficiency, we reduce the number of epochs. For the second stage follows the identical hyperparameters used in prior multi-task learning works (Ye & Xu, 2022; 2023b; Yang et al., 2024d). We implement our methodology with PyTorch (Paszke et al., 2019) and run experiments on NVIDIA H100 GPUs with 96G VRAM.

Table 11: Training hyperparameters setup.

| Hyperparameters | Stage 1 | Stage 2 |
|---|---|---|
| #GPUs | 8 | 2 |
| Total batch size | 128 | 4 |
| Optimizer | AdamW | AdamW |
| Base learning rate | 1e-3 | 2e-5 |
| LR scheduler | cosine | poly |
| Weight decay | 1e-2 | 1e-6 |
| Steps | 30 epochs | 40,000 iterations |
| Warmup | 2 epochs | 0 |
| Gradient clipping | 0 | 10 |

## C ADDITIONAL EXPERIMENTAL RESULTS

**Quantitative analysis of representation biases.** In Figure 1, we provide a quantitative analysis of the representation biases of VFMs, here we present the complete results in Table 12. The

'ST Full' and 'MT Full' results of ViT is used as the single-task and multi-task baseline in other comparisons, respectively. We also include the multi-teacher VFM distillation methods, RADIO and Theia, along with our proposed SAK for comparison. In the 'Freeze' setting of SAK, we freeze the Teacher-Agnostic Stem and Teacher-Specific Adapter Paths, and tune the Mixture-of-Representations Routers and prediction heads during the second stage. The results indicate that SAK outperforms all three VFMs and two baselines, achieving superior overall performance and more balanced improvements across the five tasks.

Table 12: Quantitative results of VFMs and multi-teacher VFM distillation baselines on PASCAL-Context. We consider three fine-tuning methods: 'ST Full' for fine-tuning the entire model in a single-task setting, 'MT Full' for fine-tuning the entire model in a multi-task setting, and 'Freeze' for freezing the backbone while training only the decoder heads, resulting in the same results for both single-task and multi-task settings.

| Model | Fine-tune | #Train Param | Semseg mIoU↑ | Parsing mIoU↑ | Saliency maxF↑ | Normal mErr↓ | Boundary odsF↑ | $\Delta_m$% ↑ |
|---|---|---|---|---|---|---|---|---|
| ViT (Dosovitskiy et al., 2021) | ST Full | 462M | 80.25 | 70.54 | 84.54 | 13.57 | 74.22 | 0.00 |
| | MT Full | 116M | 76.76 | 65.26 | 84.39 | 13.98 | 70.37 | -4.04 |
| | Freeze | 30M | 71.36 | 56.08 | 81.06 | 17.17 | 64.03 | -15.19 |
| DINOv2 (Oquab et al., 2024) | ST Full | 462M | 80.77 | 68.62 | 85.30 | 13.24 | 78.55 | 1.42 |
| | MT Full | 116M | 77.89 | 70.57 | 84.89 | 13.62 | 74.27 | -0.56 |
| | Freeze | 30M | 81.18 | 74.38 | 80.48 | 16.85 | 70.83 | -5.39 |
| CLIP (Radford et al., 2021) | ST Full | 462M | 79.83 | 67.16 | 84.65 | 13.39 | 74.93 | -0.58 |
| | MT Full | 116M | 76.84 | 65.87 | 84.61 | 13.91 | 70.50 | -3.66 |
| | Freeze | 30M | 78.33 | 65.31 | 81.48 | 16.84 | 67.43 | -9.33 |
| SAM (Kirillov et al., 2023) | ST Full | 474M | 71.13 | 68.03 | 86.52 | 13.26 | 79.51 | -0.63 |
| | MT Full | 118M | 66.39 | 65.65 | 85.38 | 13.74 | 77.20 | -4.09 |
| | Freeze | 30M | 49.11 | 59.85 | 81.10 | 16.21 | 75.89 | -15.05 |
| RADIO (Ranzinger et al., 2024b) | MT Full | 128M | 78.06 | 68.13 | 85.18 | 13.59 | 72.64 | -1.53 |
| | Freeze | 30M | 81.42 | 71.71 | 82.43 | 16.21 | 72.91 | -4.12 |
| Theia (Shang et al., 2024) | MT Full | 115M | 76.51 | 67.53 | 84.38 | 14.56 | 70.34 | -4.33 |
| | Freeze | 30M | 62.09 | 54.50 | 81.69 | 16.87 | 63.70 | -17.45 |
| SAK (Ours) | MT Full | 134M | 81.65 | 72.38 | 84.87 | 14.05 | 73.23 | **-0.03** |
| | Freeze | 36M | 80.61 | 71.15 | 83.34 | 15.22 | 71.94 | **-3.07** |

**Detailed results corresponding to Figure 3.** We also provide the details of Figure 3 in Tables 13 and 14 with additional state of the arts in multi-task learning included. SAK is distilled from VFM teachers based on ViT-L backbones. On PASCAL-Context, our approach outcomes a positive MTL Gain $\Delta_m$, surpassing the single-task baseline—a groundbreaking improvement not achieved by other models, even those with complex task interaction modules designed particularly for MTL. On both datasets, SAK not only outperforms RADIO and Theia but also surpasses the multi-task SOTAs with significant margins, all while utilizing a simple architecture and considerably fewer parameters.

Table 13: Comparison with state of the arts on PASCAL-Context, based on ViT-B backbones. For models not providing exact number of parameters in the paper or source code, we estimate their parameter counts.

| Model | Backbone | #Param | Semseg mIoU↑ | Parsing mIoU↑ | Saliency maxF↑ | Normal mErr↓ | Boundary odsF↑ | $\Delta_m$% ↑ |
|---|---|---|---|---|---|---|---|---|
| Single-task baseline | ViT-B | 462M | 80.25 | 70.54 | 84.54 | 13.57 | 74.22 | 0.00 |
| Multi-task baseline | ViT-B | 116M | 76.76 | 65.26 | 84.39 | 13.98 | 70.37 | -4.04 |
| InvPT (Ye & Xu, 2022) | ViT-B | 176M | 77.33 | 66.62 | 85.14 | 13.78 | 73.20 | -2.28 |
| InvPT++ (Ye & Xu, 2024) | ViT-B | ~176M | 76.95 | 66.89 | 85.12 | 13.54 | 73.30 | -1.92 |
| TaskPrompter (Ye & Xu, 2023b) | ViT-B | 418M | 79.00 | 67.00 | 85.05 | **13.47** | 73.50 | -1.24 |
| TaskExpert (Ye & Xu, 2023a) | ViT-B | 347M | 78.45 | 67.38 | 84.96 | 13.55 | 72.30 | -1.73 |
| BFCI (Zhang et al., 2023b) | ViT-B | 230M | 77.98 | 68.19 | 85.06 | 13.48 | 72.98 | -1.31 |
| MLoRE (Yang et al., 2024d) | ViT-B | 259M | 79.26 | 67.82 | **85.31** | 13.65 | **74.69** | -0.83 |
| RADIO (Ranzinger et al., 2024b) | ViT-B | 128M | 78.06 | 68.13 | 85.18 | 13.59 | 72.64 | -1.53 |
| Theia (Shang et al., 2024) | ViT-B | 115M | 76.51 | 67.53 | 84.38 | 14.56 | 70.34 | -4.33 |
| SAK (Ours) | ViT-B | 134M | **81.88** | **74.30** | 84.79 | 14.02 | 74.09 | **0.83** |

**Comparison with state of the arts using additional backbones.** As depicted in Tables 15 and 16, we extend our comparison to state of the arts based on the ViT-S backbones. Additionally, we include models using the Swin-S backbones, though they are substantially larger in model size compared to ours. Here SAK is distilled from VFM teachers based on ViT-B backbones. Our approach consistently delivers the best performance on both datasets, with particularly strong improvements

Table 14: Comparison with state of the arts on NYUD-v2, based on ViT-B backbones.

| Model | Backbone | #Param | Semseg mIoU↑ | Depth RMSE↓ | Normal mErr↓ | Boundary odsF↑ | $\Delta_m\%$ ↑ |
|---|---|---|---|---|---|---|---|
| Single-task baseline | ViT-B | 369M | 51.15 | 0.5792 | 19.77 | 77.35 | 0.00 |
| Multi-task baseline | ViT-B | 110M | 49.27 | 0.5823 | 19.92 | 75.88 | -1.72 |
| InvPT (Ye & Xu, 2022) | ViT-B | 161M | 50.30 | 0.5367 | 19.00 | 77.60 | 2.47 |
| InvPT++ (Ye & Xu, 2024) | ViT-B | ~161M | 49.79 | 0.5318 | 18.90 | 77.10 | 2.40 |
| TaskPrompter (Ye & Xu, 2023b) | ViT-B | 160M+ | 50.40 | 0.5402 | 18.91 | 77.60 | 2.49 |
| ECS (Shoouri et al., 2023) | ViT-B | 122M | 50.46 | 0.5332 | 18.42 | 77.89 | 3.53 |
| BFCI (Zhang et al., 2023b) | ViT-B | 200M+ | 51.14 | 0.5186 | 18.92 | 77.98 | 3.89 |
| TSP (Wang et al., 2024b) | ViT-B | 160M+ | 51.22 | 0.5301 | 18.78 | 76.90 | 3.26 |
| SEM (Huang et al., 2024a) | ViT-B | - | 51.34 | 0.5222 | 18.95 | 77.60 | 3.67 |
| RADIO (Ranzinger et al., 2024b) | ViT-B | 122M | 55.03 | 0.5186 | 18.49 | 77.97 | 6.33 |
| Theia (Shang et al., 2024) | ViT-B | 109M | 51.80 | 0.5367 | 19.70 | 76.08 | 1.83 |
| SAK (Ours) | ViT-B | 126M | **59.93** | **0.4942** | **17.60** | **78.60** | **11.11** |

in segmentation tasks. The slight underperformance in boundary detection and depth estimation can be attributed to the architectural differences between ViT and Swin Transformer (Liu et al., 2021c).

Table 15: Comparison with state of the arts on PASCAL-Context, based on ViT-S and Swin-S backbones. Note that ViT-S backbone has 22M parameters while Swin-S backbone has 50M parameters.

| Model | Backbone | #Param | Semseg mIoU↑ | Parsing mIoU↑ | Saliency maxF↑ | Normal mErr↓ | Boundary odsF↑ | $\Delta_m\%$ ↑ |
|---|---|---|---|---|---|---|---|---|
| Single-task baseline | ViT-S | 117M | 77.59 | 66.58 | 84.60 | 14.16 | 70.95 | 0.00 |
| Multi-task baseline | ViT-S | 29M | 73.94 | 61.22 | 83.96 | 14.89 | 67.03 | -4.84 |
| TaskPrompter (Ye & Xu, 2023b) | ViT-S | 132M | 76.57 | 63.22 | 84.54 | **13.93** | 70.60 | -1.06 |
| TaskExpert (Ye & Xu, 2023a) | ViT-S | 55M | 75.04 | 62.68 | 84.68 | 14.22 | 68.80 | -2.50 |
| MLoRE (Yang et al., 2024d) | ViT-S | 44M | 75.64 | 62.65 | **84.70** | 14.43 | 69.81 | -2.36 |
| MQTransformer (Xu et al., 2023a) | Swin-S | 57M+ | 71.25 | 60.11 | 84.05 | 14.74 | 71.80 | -4.29 |
| DeMT (Xu et al., 2023c) | Swin-S | 54M | 72.01 | 58.96 | 83.20 | 14.57 | 72.10 | -4.31 |
| DeMTG (Xu et al., 2023b) | Swin-S | 55M+ | 71.54 | 61.49 | 83.70 | 14.90 | 72.20 | -3.99 |
| ATMPNet (Sirejiding et al., 2024a) | Swin-S | 50M+ | 70.58 | 61.17 | 83.96 | 14.41 | 73.15 | -3.32 |
| TFUT (Xin et al., 2024b) | Swin-S | 63M+ | 72.49 | 63.24 | 84.06 | 14.42 | **73.50** | -2.09 |
| Theia (Shang et al., 2024) | ViT-S | 29M | 70.82 | 62.71 | 83.95 | 14.87 | 67.52 | -5.03 |
| SAK (Ours) | ViT-S | 34M | **78.66** | **68.46** | 84.66 | 14.33 | 70.28 | **0.43** |

Table 16: Comparison with state of the arts on NYUD-v2, based on ViT-S and Swin-S backbones.

| Model | Backbone | #Param | Semseg mIoU↑ | Depth RMSE↓ | Normal mErr↓ | Boundary odsF↑ | $\Delta_m\%$ ↑ |
|---|---|---|---|---|---|---|---|
| Single-task baseline | ViT-S | 94M | 48.11 | 0.5911 | 20.27 | 75.17 | 0.00 |
| Multi-task baseline | ViT-S | 28M | 45.87 | 0.6626 | 20.55 | 74.44 | -4.78 |
| MQTransformer (Xu et al., 2023a) | Swin-S | 57M | 49.18 | 0.5785 | 20.81 | 77.00 | 1.03 |
| DeMT (Xu et al., 2023c) | Swin-S | 53M | 51.50 | 0.5474 | 20.02 | 78.10 | 4.89 |
| DeMTG (Xu et al., 2023b) | Swin-S | 55M | 52.23 | 0.5599 | 20.05 | **78.40** | 4.81 |
| ATMPNet (Sirejiding et al., 2024a) | Swin-S | 50M+ | 51.82 | 0.5526 | 20.11 | 78.27 | 4.78 |
| TFUT (Xin et al., 2024b) | Swin-S | 63M | 50.04 | **0.5419** | 20.08 | 78.30 | 3.99 |
| Theia (Shang et al., 2024) | ViT-S | 28M | 46.54 | 0.6047 | 20.95 | 74.59 | -2.42 |
| SAK (Ours) | ViT-S | 32M | **54.30** | 0.5785 | **19.67** | 76.58 | **4.96** |

**Detailed results corresponding to Table 3.** In Table 3, we use a higher similarity between the student and teachers to validate the effectiveness of our SAK in preserving representation biases. The detailed breakdown of these similarity calculations for each VFM teacher is listed in Table 17. The substantially higher similarity scores further confirm our claim.

**Detailed results corresponding to Figure 5.** In Figure 5, we investigate the impact of downstream data size by experimenting with different percentages of samples from the PASCAL-Context dataset. From the detailed results in Table 18, we can confidently conclude that SAK is a more robust method compared to the multi-task baseline and multi-teacher VFM distillation methods. Besides the knowledge transferred from VFM teachers through the large-scale ImageNet dataset during the first stage, the more specialized knowledge distilled in the second stage further benefits SAK's robustness and generalization to diverse data sizes.

Table 17: Cosine similarity between the representations of student and corresponding teachers on the ImageNet-1k validation set.

| TSAP | DINOv2 | CLIP | SAM | Avg |
|------|--------|------|-----|-----|
| ✗ | 0.3566 | 0.5159 | 0.1306 | 0.3344 |
| ✔ | 0.8615 | 0.9138 | 0.8369 | 0.8707 |

Table 18: Performance w.r.t. different percentages of downstream dataset. MTL Gain is computed w.r.t. single-task baseline trained on full dataset.

| Data Percentage | Model | Semseg mIoU↑ | Parsing mIoU↑ | Saliency maxF↑ | Normal mErr↓ | Boundary odsF↑ | $\Delta_m\%\uparrow$ |
|-----------------|-------|--------------|---------------|----------------|--------------|----------------|----------------------|
| 25% | Multi-task baseline | 72.10 | 59.71 | 81.40 | 15.63 | 68.87 | -10.32 |
| | RADIO (Ranzinger et al., 2024b) | 73.47 | 63.18 | **81.94** | **15.13** | 71.49 | -7.43 |
| | Theia (Shang et al., 2024) | 71.12 | 63.65 | 81.73 | 15.57 | 69.18 | -9.20 |
| | SAK (Ours) | **80.09** | **69.77** | 81.47 | 15.28 | **72.32** | **-4.02** |
| 50% | Multi-task baseline | 74.95 | 60.65 | 83.04 | 14.72 | 69.59 | -7.42 |
| | RADIO (Ranzinger et al., 2024b) | 76.23 | 63.90 | **83.67** | **14.27** | 72.07 | -4.70 |
| | Theia (Shang et al., 2024) | 74.60 | 64.01 | 83.19 | 14.89 | 69.84 | -6.70 |
| | SAK (Ours) | **81.33** | **70.19** | 83.35 | 14.54 | **72.71** | **-1.95** |
| 75% | Multi-task baseline | 75.17 | 61.23 | 84.08 | 14.15 | 70.04 | -6.00 |
| | RADIO (Ranzinger et al., 2024b) | 76.82 | 64.26 | **84.59** | **13.76** | 72.29 | -3.42 |
| | Theia (Shang et al., 2024) | 74.98 | 64.57 | 83.83 | 14.64 | 70.04 | -5.88 |
| | SAK (Ours) | **81.32** | **70.51** | 84.33 | 14.28 | **73.01** | **-1.16** |
| 100% | Multi-task baseline | 76.76 | 65.26 | 84.39 | 13.98 | 70.37 | -4.04 |
| | RADIO (Ranzinger et al., 2024b) | 78.06 | 68.13 | **85.18** | **13.59** | 72.64 | -1.53 |
| | Theia (Shang et al., 2024) | 76.51 | 67.53 | 84.38 | 14.56 | 70.34 | -4.33 |
| | SAK (Ours) | **81.65** | **72.38** | 84.87 | 14.05 | **73.23** | **-0.03** |

**Upper-bound results of teacher amalgamation.** To evaluate the upper-bound performance of amalgamating the VFM teachers, we build an upper-bound model using the frozen encoders of three VFM teachers—DINOv2, CLIP, and SAM—along with a learnable ViT encoder as a surrogate for the Teacher-Agnostic Stem (TAS). All components employ a ViT-B backbone. For fusing the representations, we explore three strategies: element-wise addition, channel concatenation, and our proposed Mixture-of-Representations (MoR).

Table 19: Comparison with upper-bound models on PASCAL-Context, based on ViT-B backbones.

| Model | Fuse | #Param | MACs | Semseg mIoU↑ | Parsing mIoU↑ | Saliency maxF↑ | Normal mErr↓ | Boundary odsF↑ | $\Delta_m\%\uparrow$ |
|-------|------|--------|------|--------------|---------------|----------------|--------------|----------------|----------------------|
| Upper-bound | Addition | 378M | 1091G | 80.27 | 71.44 | 84.74 | **13.82** | 72.69 | -0.47 |
| | Concat | 820M | 7661G | **82.26** | **73.85** | 84.63 | 13.98 | **75.55** | **1.21** |
| | MoR | 384M | 1097G | 80.58 | 72.73 | **84.88** | 14.05 | 74.01 | 0.02 |
| SAK (Ours) | MoR | **134M** | **544G** | 81.65 | 72.38 | 84.87 | 14.05 | 73.23 | -0.03 |

Table 19 shows that a naive addition of representations fails to fully leverage the teachers' knowledge, as mentioned in the introduction and further supported by our ablation study in Section 4.3 and Table 3. While channel concatenation mitigates this limitation and leads to upper-bound performance, it comes at the cost of a significant increase in parameter count and computational overhead due to the expanded dimensionality of the fused representations. In contrast, our MoR approach surpasses addition by 0.5% while maintaining comparable parameters and MACs, thereby demonstrating its effectiveness even in this alternative setup.

Compared to these upper-bound models, our SAK outperforms the naive addition with around 1/3 of the parameters and half of the computational cost. This further validates the effectiveness and efficiency of our method.

**Balancing distillation loss and task losses.** In the second training stage, we apply a combination of distillation loss and task-specific losses, weighted by the balancing factor $\gamma$ introduced in Equation 6. We explore the impact of this factor on the final prediction results in Table 20, using teachers and student based on ViT-B backbones. As the balancing factor increases, We observe raising accuracy in segmentation and boundary detection tasks, while performance in saliency detection and surface normal estimation diminishes. This suggests that segmentation and boundary detection rely heavily

on knowledge distilled from the VFM teachers, whereas the other two tasks depend more on domain-specific details provided by downstream supervision, resulting in a seesaw phenomenon. Though a factor of 2.0 yields the best overall performance, we use a default value of 1.0 for simplicity.

Table 20: Performance w.r.t. different values of balancing factor $\gamma$.

| $\gamma$ | Semseg mIoU↑ | Parsing mIoU↑ | Saliency maxF↑ | Normal mErr↓ | Boundary odsF↑ | $\Delta_m\%$ ↑ |
|---|---|---|---|---|---|---|
| 0.5 | 81.19 | 71.98 | **84.95** | **14.01** | 73.06 | -0.22 |
| 0.75 | 81.46 | 72.22 | 84.91 | 14.03 | 73.17 | -0.10 |
| 1.0 | 81.65 | 72.38 | 84.87 | 14.05 | 73.23 | -0.03 |
| 1.25 | 81.76 | 72.51 | 84.84 | 14.06 | 73.29 | 0.03 |
| 1.5 | 81.84 | 72.62 | 84.81 | 14.08 | 73.33 | 0.06 |
| 2.0 | 81.92 | 72.78 | 84.74 | 14.11 | 73.37 | **0.07** |
| 2.5 | 81.97 | 72.88 | 84.68 | 14.14 | **73.39** | 0.06 |
| 3.0 | 81.99 | 72.95 | 84.62 | 14.17 | **73.39** | 0.03 |
| 4.0 | **82.01** | **73.01** | 84.53 | 14.22 | 73.38 | -0.05 |

**Gating weights learned by MoR Routers.** We present the complete version of Figure 4 in Figure 7, along with the results from NYUD-v2 in Figure 8. A similar observation across both datasets is that all tasks tend to depend on the shared TAS representations at lower levels. Conversely, at higher levels, the gating weights for each task diverge, showing varied dependencies on different experts. This demonstrate that the biases from different VFM teachers are adaptively leveraged by the tasks, allowing the model to optimize performance through selective representation fusion.

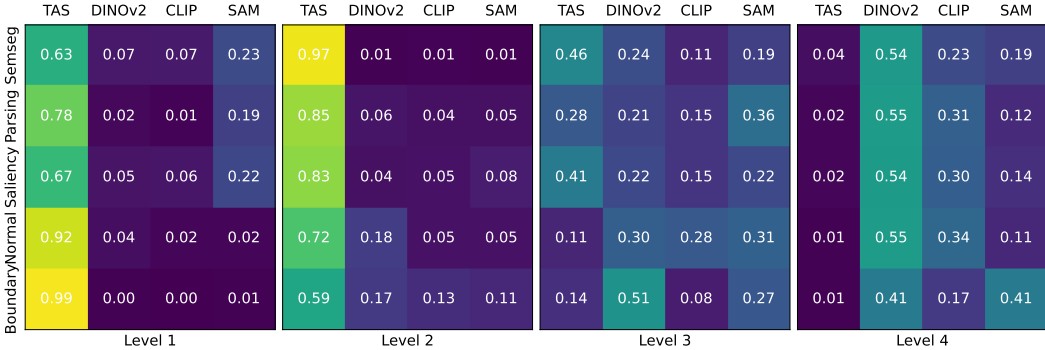

Figure 7: Gating weights of different experts learned by MoR Routers on PASCAL-Context.

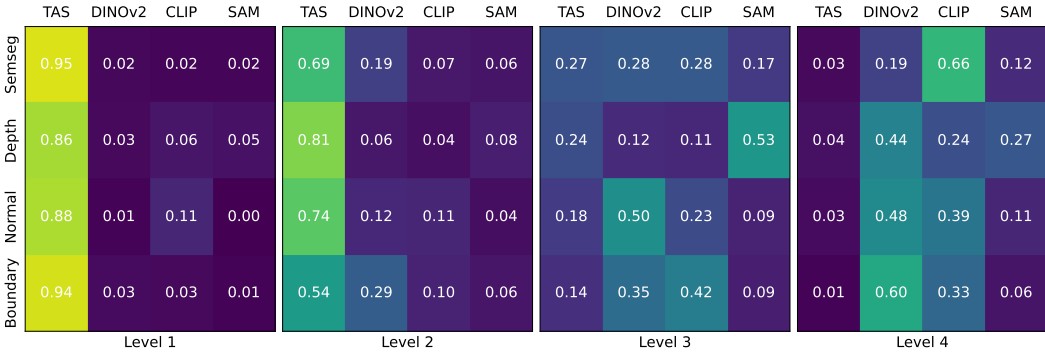

Figure 8: Gating weights of different experts learned by MoR Routers on NYUD-v2.

# D ADDITIONAL DISCUSSIONS

## D.1 MODEL EFFICIENCY

**Training cost.** Since SAK employs a multi-teacher distillation framework, the iterative forwarding of VFM teachers inevitably increases training costs, which cannot be overlooked. However, this process is a common requirement for all multi-teacher distillation methods, and when compared to other baselines, such as RADIO, SAK demonstrates remarkable efficiency.

In Table 21, we calculate the number of parameters and computational costs for teachers and students in RADIO and SAK, both using a ViT-L backbone for the student. The results show that RADIO incurs 2.5 times the parameters and forwarding costs of SAK. Additionally, RADIO's larger input size of 1,024 introduces more tokens in the representations, further reducing the backward efficiency. Hence, SAK shows significantly lower demands in RAM, GPU memory, training time, and storage.

Table 21: Comparison with RADIO on number of parameters and computational cost (measured by MACs, Multiply–accumulate operations) of all teachers and students. The calculation is based on students with ViT-L backbones for both methods. SAK uses VFM teachers with ViT-L backbones.

| RADIO teachers and students | Input size | #Param | MACs | SAK teachers and students | Input size | #Param | MACs |
|---|---|---|---|---|---|---|---|
| DFN CLIP-H/14 | 378 | 631M | 460G | DINOv2-L/14 | 384 | 304M | 221G |
| SigLIP-SO400M/14 | 384 | 413M | 300G | OpenCLIP-L/14 | 384 | 304M | 221G |
| DINOv2-g/14 | 224 | 1135M | 291G | SAM-L/16 | 1024 | 307M | 1308G |
| SAM-H/16 | 1024 | 636M | 2730G | | | | |
| RADIOv2.5-L/16 | 1024 | 320M | 1240G | SAK-L/16 | 384 | 343M | 198G |
| Sum over all teachers and student | | **3135M** | **5021G** | Sum over all teachers and student | | **1258M** | **1948G** |

Moreover, as listed in Table 29, RADIO is trained on the large-scale DataComp-1B dataset (Gadre et al., 2023) (1.4B images) with 614M total samples seen. In contrast, SAK is trained on ImageNet-1k dataset (1.2M images) with only 36M samples seen, further highlighting the efficiency of SAK in training and memory costs.

**Inference cost.** It is important to note that our framework does not require teachers during inference. Instead, the representations distilled from the teachers are generated directly by the lightweight TSAP modules, which introduce minimal additional parameters and computational overhead. To provide a clearer perspective, we present the number of parameters and computational cost introduced by our modules when integrated into the ViT-B or ViT-L backbone. Table 22 indicates that each TSAP branch accounts for less than 5% of the backbone's parameters and computations (4M *vs.* 86M parameters, 4G *vs.* 88G MACs with ViT-B). Similarly, each MoR Router is as lightweight as 2M parameters and 2G MACs, even with the larger ViT-L backbone.

Table 22: Number of parameters and computational cost when integrating different components into SAK, based on ViT-B or ViT-L backbone. The calculation is based on a student corresponding to three teachers on the five-task PASCAL-Context dataset.

| Model | #Param | MACs | Model | #Param | MACs |
|---|---|---|---|---|---|
| ViT-B/16 | 86M | 88G | ViT-L/16 | 304M | 311G |
| +TSAP (3 branches) | 98M | 100G | +TSAP (3 branches) | 344M | 351G |
| +MoR (5 tasks) | 104M | 106G | +MoR (5 tasks) | 354M | 362G |

## D.2 DIFFERENCES BETWEEN MOE AND MOR

Unlike Mixture-of-Experts (MoE) used in previous multi-task learning (Yang et al., 2024d; Chen et al., 2023; Liang et al., 2022; Ye & Xu, 2023a) or general-purpose models (Shazeer et al., 2017; Jacobs et al., 1991), where the experts are typically homogeneous during optimization, the experts in our Mixture-of-Representation (MoR) approach consist of representations distilled from various VFM teachers as well as the TAS. These experts inherently possess different knowledge, making them heterogeneous by design. As a result, the MoR Router does not require a balanced routing because task-specific routers dynamically weigh and combine the most relevant representations for each task. Empirically, we do not encounter any difficulties in training the MoR Routers. This approach ensures that the router effectively synergizes the complementary strengths of the diverse representations to optimize performance for different tasks.

## D.3 VFM Teacher Selection

**Principles of VFM teacher selection.** It is well-established that DINOv2, CLIP, and SAM are among the most widely used Vision Foundation Models. Prior baselines RADIO and Theia also employ them for experiments. Besides, RADIO uses DFN CLIP (Fang et al., 2024) and SigLIP (Zhai et al., 2023), both of which are improved models derived from CLIP. However, as shown in the Table 23, DFN CLIP and SigLIP fail to attain comparable outcomes to CLIP. For this reason, we choose to adhere to CLIP when selecting teachers.

Table 23: Comparison of CLIP, DFN CLIP, and SigLIP on PASCAL-Context, based on ViT-B backbones.

| Model | Fine-tune | Semseg mIoU↑ | Parsing mIoU↑ | Saliency maxF↑ | Normal mErr↓ | Boundary odsF↑ | $\Delta_m\%$ ↑ |
|---|---|---|---|---|---|---|---|
| CLIP (Radford et al., 2021) | MT Full | 76.84 | 65.87 | 84.61 | 13.91 | 70.50 | **-3.66** |
| | Freeze | 78.33 | 65.31 | 81.48 | 16.84 | 67.43 | **-9.33** |
| DFN CLIP (Fang et al., 2024) | MT Full | 75.19 | 63.78 | 84.63 | 14.00 | 70.57 | -4.77 |
| | Freeze | 69.35 | 53.48 | 81.36 | 17.24 | 64.85 | -16.24 |
| SigLIP (Zhai et al., 2023) | MT Full | 75.32 | 63.57 | 84.67 | 14.00 | 70.50 | -4.81 |
| | Freeze | 51.72 | 45.34 | 77.93 | 18.35 | 60.33 | -26.61 |

Theia utilizes ViT (pretrained on ImageNet) and Depth Anything (Yang et al., 2024b) as additional teachers. Since the Teacher-Agnostic Stem in SAK is initialized with an ImageNet-pretrained ViT backbone, we do not include ViT as an additional teacher to reduce training costs. While Depth Anything is a high-impact VFM, we leave its exploration for future work.

Additionally, we avoid using very large models, *e.g.*, ViT-H or ViT-g based, in our experiments due to constraints in computational resources (RADIO uses 64 GPUs, while we can only use 8 GPUs). It is worth noting that SAK is a highly flexible framework, capable of integrating and benefiting further from more powerful VFM teachers, making it adaptable to various settings.

**Sensitivity of SAK to VFM teacher selection.** We supplement another analysis with a SAK student distilled from DINOv2 and SigLIP teachers, with ViT-B backbones for both teachers and student. As shown in Table 24, compared with the student distilled from DINOv2 and CLIP teachers (presented in Table 7), it exhibits better performance on three out of five tasks, while achieving comparable overall accuracy, further underscoring the robustness and adaptability of our method.

Table 24: Performance w.r.t. different combinations of VFM teachers on PASCAL-Context.

| Teachers | Semseg mIoU↑ | Parsing mIoU↑ | Saliency maxF↑ | Normal mErr↓ | Boundary odsF↑ | $\Delta_m\%$ ↑ |
|---|---|---|---|---|---|---|
| DINOv2+CLIP | **81.53** | **71.95** | 84.49 | 14.16 | 72.59 | **-0.60** |
| DINOv2+SigLIP | 81.41 | 71.03 | **84.62** | **14.06** | **72.92** | -0.63 |

## D.4 Framework Extension

As highlighted in our introduction, SAK is a highly flexible framework that can be easily adjusted to add new teachers or new downstream tasks.

**Adding new VFM teachers.** When adding a new teacher, we freeze the existing TAS and TSAP modules to preserve the knowledge transferred from the previous teachers A new TSAP module is then introduced and distilled for the new teacher, facilitating efficient adaptation. In the second stage, we train on the downstream dataset with all VFM teachers and tune the entire student model.

We conduct an experiment by incorporating a new SigLIP teacher into a student already distilled from DINOv2, CLIP, and SAM. As shown in Table 25, including SigLIP teacher boosts performance on Semseg and Normal tasks and yields competitive results on Saliency and Boundary tasks. However, since SigLIP encodes knowledge similar to CLIP during pretraining and does not surpass CLIP on the benchmark, it is reasonable that it could not lead to further improvements. Moreover, the additional representations may increase the difficulty for the MoR Routers in learning optimal weights, potentially accounting for the degradation in human parsing.

Table 25: Performance of SAK with additional SigLIP teacher on PASCAL-Context.

| Teachers | Semseg mIoU↑ | Parsing mIoU↑ | Saliency maxF↑ | Normal mErr↓ | Boundary odsF↑ | $\Delta_m\%$ ↑ |
|---|---|---|---|---|---|---|
| DINOv2+CLIP+SAM | 81.65 | **72.38** | **84.87** | 14.05 | **73.23** | **-0.03** |
| +SigLIP | **81.78** | 70.66 | 84.85 | **14.02** | 73.21 | -0.45 |

**Adding different tasks.** Additionally, when applying to different downstream tasks, it is only necessary to add task-specific decoders and perform the second-stage training (with distillation) or fine-tuning (without distillation), as the first-stage distillation is agnostic to the downstream tasks. This process is significantly more efficient than the first stage. Meanwhile, we can readily remove a branch of TSAP if it is no longer necessary, without affecting the functionality of the TAS or other TSAP branches, ensuring the modularity and adaptability of our design. We have also shown in Figure 5 and its accompanying analysis that SAK exhibits strong robustness and generalization capability in downstream tasks.

## D.5 COMPARISON OF SAK WITH TEACHERS

**Why cannot SAK outperform teachers at every task?** Firstly, performance loss is inevitable during distillation, since our distillation process is lightweight compared to the extensive pretraining of VFM teachers. As detailed in Table 1, our VFM teachers are pretrained on large-scale datasets containing hundreds to thousands of times more image samples than the ImageNet-1k dataset used for distillation. Moreover, the Teacher-Specific Adapter Path, designed to preserve teachers' biases, utilizes a standard and simple architecture with less than 5% of the backbone's parameters and computations. Consequently, it is reasonable that the distilled SAK cannot fully inherit teachers' knowledge when compressing three teacher models into a single student of similar model size, accounting for why our approach cannot outperform the teachers on their proficient tasks. We believe that leveraging larger datasets could further enhance knowledge transfer during distillation and benefit downstream tasks. Another reason is that SAM operates at a higher input resolution of 1,024, rather than 512 for other models, which bolsters its strength in tasks like boundary detection that rely heavily on fine image details.

**Why can SAK outperform teachers on specific tasks?** The observed superiority of SAK over teachers stems from our motivation to synergize the complementary strengths of teachers. Complementarity exists not only across tasks but also within tasks. Take semantic segmentation as an example, as illustrated in Figure 1 and qualitative analysis in Section 2.2, DINOv2 excels at capturing localized features but CLIP offers strong object-level understanding with its rich semantic knowledge from the language domain. SAM produces exceptional fine-grained pixel-level masks due to its higher resolution. Additional analysis of representations in Appendix A further supports these observations. By preserving the intra-task representation biases during distillation and amalgamating them using proposed Mixture-of-Representations, SAK achieves improved performance.

Another contributing factor is the balance between common knowledge and task-specific information. Regarding saliency estimation and surface normal estimation, three VFM teachers perform suboptimally when frozen, as they lack downstream task-specific information. Though fine-tuning can alleviate this limitation, it results in degradation of accuracy in segmentation tasks for DINOv2 and CLIP, as shown in the Table 12. This trade-off, known as the negative transfer problem in multi-task learning (Vandenhende et al., 2021; 2020), highlights the challenge of balancing task-specific and pretrained knowledge.

In contrast, SAK preserves the representation biases of the frozen teachers within the TSAP modules while leveraging the TAS to learn shared knowledge and downstream information. This disentanglement ensures knowledge diversity without mutual interference. Moreover, MoR Routers dynamically weigh and combine the most relevant representations for each task, bridging the gap between general-purpose knowledge and task-specific characteristics.

### D.6 EVALUATION ON ADDITIONAL TASKS

First, we emphasize that our work is less targeted at single-task learning. Instead, our key motivation lies in the inherent representation biases of VFMs, which result in advantages and disadvantages across different vision tasks. Our SAK synergizes these complementary biases to enhance multi-task learning. Therefore, the tasks we evaluated are defined by the datasets, since each image sample must be labeled for every task included. Nevertheless, we also provide additional evaluations on single-task learning, specifically for classification, depth estimation, and vision-language learning, to offer a more comprehensive analysis.

**Linear classification.** Following the setup of DINOv2, we evaluate linear probing on the ImageNet-1k dataset. We freeze the backbones and train a linear classifier and the MoR Routers of SAK using the same hyperparameters as DINOv2. For comparison, we also evaluate the VFM teachers and RADIO, while reporting Theia's result from its original paper.

Table 26 shows that SAK outperforms the two baselines, despite a performance gap compared to the DINOv2 and CLIP teachers due to inevitable loss during distillation. It is worth noting that RADIO uses more powerful teachers, including DINOv2-g/14-reg (accuracy 87.1), while Theia uses DINOv2-L/14 teacher (accuracy 86.3).

Table 26: Comparison with VFMs and baselines on linear classification on ImageNet-1k, reported by Top-1 accuracy on the validation set. We freeze the backbone and perform linear probing at resolution of 224.

| Model | Backbone | Accuracy↑ |
|---|---|---|
| DINOv2 (Oquab et al., 2024) | ViT-B/14 | 84.5 |
| CLIP (Radford et al., 2021) | ViT-B/16 | **84.7** |
| SAM (Kirillov et al., 2023) | ViT-B/16 | 46.9 |
| RADIO (Ranzinger et al., 2024b) | ViT-B/16 | 78.2 |
| Theia (Shang et al., 2024) | ViT-B/16 | 75.2 |
| SAK (Ours) | ViT-B/16 | 79.1 |

**Depth estimation.** We also evaluate the monocular depth estimation task on the NYUd dataset (Silberman et al., 2012) (note it is different from the NYUDv2 dataset used in our primary experiments), which is the proficient task of DINOv2. We follow the evaluation protocol (Li et al., 2024b) used by DINOv2 and use the identical experimental setups and decode heads (lin. 1 and lin. 4). Since DINOv2 does not provide a complete codebase for this task, we reproduce the pipeline. We also evaluate ViT, SAM, Theia, and include baselines reported in the DINOv2 paper, namely OpenCLIP (Cherti et al., 2023), MAE (He et al., 2022), and DINO (Caron et al., 2021).

As demonstrated in Table 27, while SAK does not fully match the teacher's performance due to inevitable loss during distillation and different patch size, it surpasses other foundation models and baselines, even those with larger backbones. Moreover, the effectiveness of synergizing multiple VFM teachers is clearly evidenced by the results, as distillation from three teachers improves SAK's performance, bringing it closer to the upper bound of the teacher.

Table 27: Comparison with VFMs and baselines on monocular depth estimation on NYUd. We freeze the backbone and perform linear probing on top of one (lin. 1) or four (lin. 4) layers.

| Model | Backbone | lin. 1 RMSE ↓ | lin. 4 RMSE ↓ |
|---|---|---|---|
| ViT (Dosovitskiy et al., 2021) | ViT-B/16 | 1.118 | 1.117 |
| SAM (Kirillov et al., 2023) | ViT-B/16 | 0.678 | 0.652 |
| Theia (Shang et al., 2024) | ViT-B/16 | 0.644 | 0.629 |
| OpenCLIP (Cherti et al., 2023) | ViT-G/14 | 0.541 | 0.510 |
| MAE (He et al., 2022) | ViT-H/14 | 0.517 | 0.483 |
| DINO (Caron et al., 2021) | ViT-B/8 | 0.555 | 0.539 |
| DINOv2 (Teacher upper bound) | ViT-B/14 | **0.399** | **0.362** |
| DINOv2 (Our reproduced) | ViT-B/14 | 0.406 | 0.366 |
| SAK (Distilled from DINOv2) | ViT-B/16 | 0.482 | 0.463 |
| SAK (Distilled from 3 teachers) | ViT-B/16 | 0.450 | 0.436 |

**Vision-language learning.** To evaluate the performance of multi-modal learning, we follow the setup of RADIO (Ranzinger et al., 2024b) and integrate SAK into LLaVA-1.5 (Liu et al., 2024b;a). We freeze the Teacher-Agnostic Stem and Teacher-Specific Adapter Path, and fine-tune only the MoR Router. We evaluate three VQA tasks – GQA, POPE, and VQAv2 – at resolutions of 432 and 512, consistent with RADIO. We also include DINOv2-L, CLIP-L, and SAM-H for comparison.

Table 28 shows that while SAK is trained on significantly less data, and with smaller VFM teachers, it achieves comparable performance to the teachers and RADIO on GQA and POPE. The performance gap on VQAv2 is largely attributable to the simplicity of our distillation setup, which prioritizes training efficiency. Regarding dataset scale, as listed in Table 1 and Table 29, all VFM teachers and RADIO use large-scale datasets, *e.g.*, DINOv2 uses 142M images, CLIP uses 400M, RADIO uses 1.4B. In contrast, our SAK distillation relies on ImageNet-1k, which has only 1.2M images. This limited dataset diversity could significantly impact performance on VQA tasks, which require extensive training data.

Table 28: Comparison with VFMs and RADIO on vision-language learning, based on LLaVA-1.5 (Liu et al., 2024a) with GQA, POPE, and VQAv2 datasets.

| Model | Backbone | Resolution | GQA | POPE | VQAv2 |
|---|---|---|---|---|---|
| DINOv2 (Oquab et al., 2024) | ViT-L/14 | 336 | 62.11 | 87.72 | 76.42 |
| CLIP (Radford et al., 2021) | ViT-L/14 | 336 | 62.20 | 86.09 | 78.49 |
| SAM (Kirillov et al., 2023) | ViT-H/16 | 1024 | 49.92 | 81.76 | 57.65 |
| RADIO (Ranzinger et al., 2024b) | ViT-B/16 | 432 | 62.09 | 85.87 | 77.24 |
| | | 512 | 62.70 | 86.59 | 78.03 |
| | ViT-L/16 | 432 | 62.89 | 86.13 | 79.44 |
| | | 512 | 63.58 | 86.66 | 80.04 |
| SAK (Ours) | ViT-B/16 | 432 | 60.84 | 85.50 | 72.80 |
| | | 512 | 60.75 | 85.84 | 74.10 |
| | ViT-L/16 | 432 | 62.01 | 86.03 | 75.31 |
| | | 512 | 62.32 | 86.75 | 75.48 |

Meanwhile, RADIO employs larger VFM teachers, including DFN CLIP-H, SigLIP-SO400M, DINOv2-g, and SAM-H, which result in 2.5 times the parameters and forwarding costs of SAK with teachers based on ViT-L, shown in Table 21. RADIO also uses a larger resolution of 1,024 during its distillation (in LLaVA experiments, the image resolution is 432 or 512), enhancing image information extraction but at a substantial increase in forwarding and backwarding costs. Considering these factors, it is reasonable that SAK cannot outperform RADIO in VLM applications. We believe that with access to larger-scale datasets and more powerful teachers during distillation, SAK could achieve competitive or superior results.

# E ADDITIONAL RELATED WORK

## E.1 KNOWLEDGE DISTILLATION

Knowledge Distillation (KD) is a model compression technique where a smaller student model learns from a larger teacher model, typically using the teacher's output logits as soft targets to guide training (Bucilă et al., 2006; Hinton et al., 2014; Ba & Caruana, 2014; Beyer et al., 2022). To improve knowledge transfer, some studies have proposed using intermediate representations (Romero et al., 2015; Ahn et al., 2019; Heo et al., 2019; Zagoruyko & Komodakis, 2017). Multi-teacher knowledge distillation has also been explored, where knowledge is transferred from multiple teacher models simultaneously or progressively to improve the student's generalization (Sau & Balasubramanian, 2016; You et al., 2017; Fukuda et al., 2017; Wen et al., 2024; Cao et al., 2023; Roth et al., 2024). Another branch of work, known as knowledge amalgamation, combines knowledge from multiple teachers tackling different tasks into a student that learns the union of all tasks (Shen et al., 2019a;b; Luo et al., 2019; Ye et al., 2019b; Vongkulbhisal et al., 2019; Thadajarassiri et al., 2021; 2023). Such methods mainly unify classification tasks from multiple domains. In contrast, our approach addresses multiple heterogeneous vision tasks (*e.g.*, semantic segmentation, surface normal estimation) simultaneously.

## E.2 Vision Foundation Models

Vision Foundation Models (VFMs) are large-scale, general-purpose vision models trained on massive datasets, demonstrating exceptional performance across a wide range of downstream tasks, particularly when generalizing to unseen tasks and domains. Some VFMs are trained with specific tasks, such as ViT (Dosovitskiy et al., 2021), DeiT (Touvron et al., 2021), and Swin Transformer (Liu et al., 2021c) for image classification, Segment Anything Model (SAM) (Kirillov et al., 2023; Ravi et al., 2025) for promptable segmentation, and Depth Anything (Yang et al., 2024b;c) for monocular depth estimation. Others are trained with pretext tasks agnostic to downstream tasks. For instance, CLIP (Radford et al., 2021) and its derivatives (Zhai et al., 2023) align features of image-text pairs, DINOv2 (Oquab et al., 2024) employs self-supervised learning, and Pang et al. (2024) even uses Large Language Model (LLM) as a visual encoder. These models have shown strong capabilities in diverse visual tasks, including low-shot classification, open-vocabulary recognition, semantic segmentation, and visual question answering (VQA).

Previous studies have reported similar conclusions as our studied representation biases when evaluating VFMs on different tasks (Ranzinger et al., 2024b; Zong et al., 2024; Kar et al., 2024; Tong et al., 2024a;b). Therefore, several existing works have sought to leverage multiple VFMs to enhance downstream performance. Typically, they extract image features from multiple VFMs and then concatenate or fuse these features (Lin et al., 2023; Kar et al., 2024; Zong et al., 2024; Tong et al., 2024a;b; Man et al., 2024; Shlapentokh-Rothman et al., 2024). While this approach can improve visual encoding capability, it comes with a major drawback: running inference across multiple vision encoders drastically increases computational costs, as well as the memory and storage requirements due to large-scale parameters. Additionally, Zong et al. (2024) highlights that biased information from VFMs can lead to performance degradation when using a simple fusion method.

## E.3 Knowledge Distillation of Vision Foundation Models

Due to their large sizes and extensive training data requirements, vision foundation models are typically computationally intensive to train from scratch. Therefore, distilling VFMs into smaller models with knowledge distillation techniques has become a popular topic (Liu et al., 2024c; Vemulapalli et al., 2024; Sun et al., 2023; Yang et al., 2024a; Zhang et al., 2024a). For example, several works (Zhang et al., 2023a; Zhou et al., 2023; Zhang et al., 2024b; Wang et al., 2023; Songa et al., 2024) have derived lightweight versions of SAM. More recently, research has explored distilling multiple VFMs into a single student model (Ranzinger et al., 2024b; Shang et al., 2024; Sariyildiz et al., 2024; Ranzinger et al., 2024a), which is closely related to our approach. Besides the differences already discussed in the main paper, we further provide a detailed comparison in Table 29. RADIO uses strong teachers with substantially larger scales and capacities, for instance, DINOv2-g has 1,100M parameters. Theia also uses a ViT-H teacher with 632M parameters, double the size of ViT-L. Moreover, RADIO is trained on a dataset over 1,000 times larger than ImageNet-1k, even with larger input resolutions and a greater number of total samples seen. Despite the relatively lower training cost, our proposed SAK demonstrates superior outcomes on downstream tasks, exceeding both baselines by notable margins.

Table 29: Comparison with multi-teacher VFM distillation methods w.r.t. training paradigms and the downstream MTL Gain $\Delta_m$ on the PASCAL-Context and NYUD-v2 dataset, respectively. All student models use ViT-B backbones and the same decoder heads.

| Model | RADIO (Ranzinger et al., 2024b) | Theia (Shang et al., 2024) | SAK (Ours) |
| --- | --- | --- | --- |
| Teachers | DFN CLIP-H, SigLIP-SO400M, DINOv2-g, SAM-H | ViT-H, DINOv2-L, CLIP-L | DINOv2-L, OpenCLIP-L, SAM-L |
| Dataset | DataComp-1B | ImageNet-1k | ImageNet-1k |
| Dataset Size | 1.4B | 1.2M | 1.2M |
| Resolution | 768 | 224 | 384 |
| #Samples | 614M | 60M | 36M |
| $\Delta_m\%$ | -1.53, 6.33 | -4.33, 1.83 | 0.83, 11.11 |

### E.4    MULTI-TASK LEARNING

Multi-Task Learning (MTL) aims to train a single model to learn multiple tasks simultaneously (Caruana, 1997; Ruder, 2017; Crawshaw, 2020; Zhang & Yang, 2021; Yu et al., 2024). MTL research primarily falls into two categories: multi-task optimization (Kendall et al., 2018; Liu et al., 2021b; Ye et al., 2021; Chen et al., 2018; Yu et al., 2020; Wang et al., 2021; Liu et al., 2021a; Javaloy & Valera, 2022; Chen et al., 2020; Sener & Koltun, 2018; Guo et al., 2018; Lu et al., 2024a) and model architecture design (Ruder et al., 2019; Long et al., 2017; Huang et al., 2024b; Wallingford et al., 2022; Mallya et al., 2018; Huang et al., 2018; Rosenbaum et al., 2018; Meyerson & Miikkulainen, 2018; Yang & Hospedales, 2017; Bragman et al., 2019).

In multi-task dense prediction, most works focus on designing architecture, which can be further divided into encoder-focused and decoder-focused methods (Vandenhende et al., 2021). Encoder-focused methods develop shared encoders to extract features for different tasks, employing techniques like feature fusion (Misra et al., 2016; Ruder et al., 2019; Gao et al., 2019), attention (Liu et al., 2019), branched networks (Brüggemann et al., 2020; Guo et al., 2020; Vandenhende et al., 2019; Lu et al., 2017; Raychaudhuri et al., 2022), and mixture-of-experts (Liang et al., 2022; Chen et al., 2023; Ye & Xu, 2023a). Decoder-focused methods, on the other hand, design complicated decoders to extract task-specific features by modeling task interactions (Xu et al., 2018; Zhang et al., 2019; Zhou et al., 2020; Zhang et al., 2018; Brüggemann et al., 2021; Vandenhende et al., 2020; Ye & Xu, 2024; Xu et al., 2023c;b; Ye & Xu, 2023b; Zhang et al., 2021; Xu et al., 2022; Sirejiding et al., 2023; Xu et al., 2023a; Sirejiding et al., 2024b; Xin et al., 2024b; Sirejiding et al., 2024a; Lin et al., 2024; Huang et al., 2024a; Zhang et al., 2023b; Wang et al., 2024b). Additional topics include task-conditional models (Maninis et al., 2019; Kanakis et al., 2020; Sun et al., 2021; Lu et al., 2024c;b), efficient adaptation (Liu et al., 2022; Xin et al., 2024a; Agiza et al., 2024; Wang et al., 2024c), efficient computation (Neseem et al., 2023; Shoouri et al., 2023; Aich et al., 2023), regularization (Yang et al., 2023; Li et al., 2024a), and generative modeling (Bao et al., 2022; Qiu et al., 2024).

## F    ADDITIONAL QUALITATIVE RESULTS

To provide an intuitive comparison between the proposed SAK, VFM teachers, and existing methods, we visualize the task predictions of frozen VFM teachers, RADIO, Theia, and our model with examples from PASCAL-Context and NYUD-v2, as shown in Figures 9, 10 (the same image used in Figure 1), and 11. All methods use ViT-B backbones, with RADIO and Theia fully fine-tuned on the downstream dataset to ensure a fair comparison. Our model noticeably generates better details and fewer errors, especially in semantic segmentation, saliency estimation, and depth estimation.

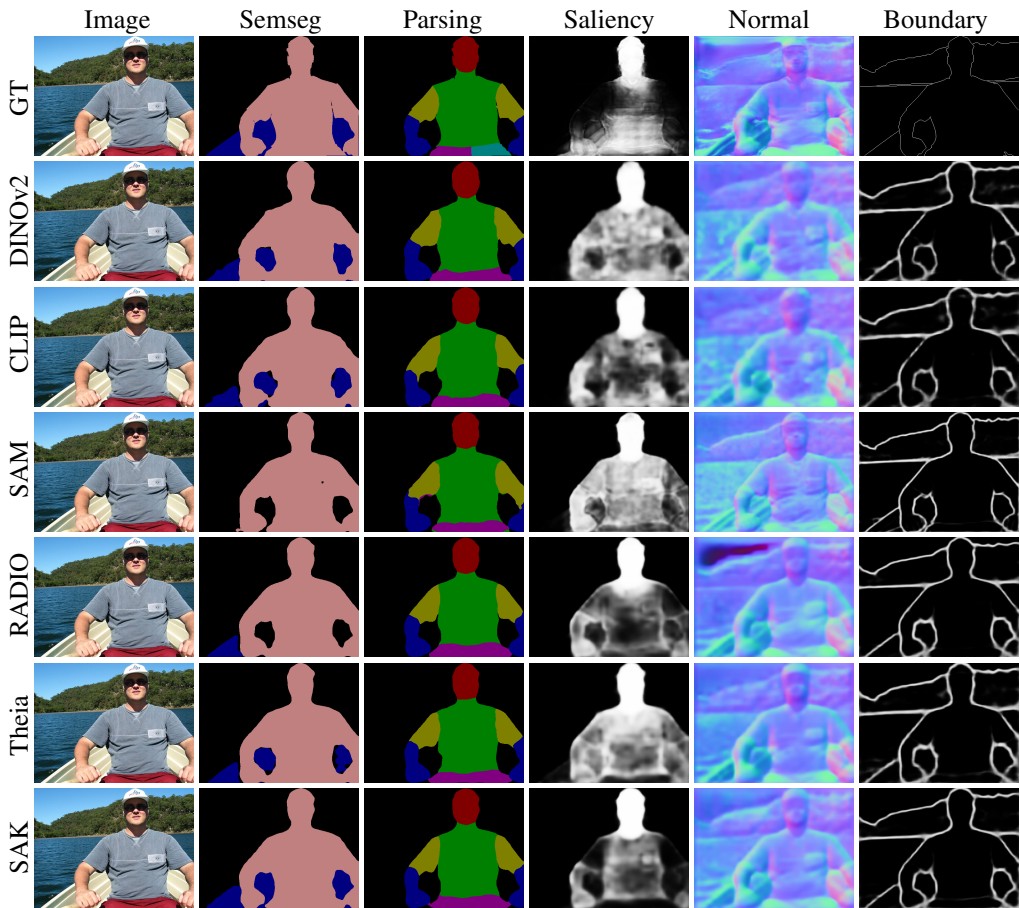

Figure 9: Qualitative results compared with VFM teachers and distillation methods on PASCAL-Context.

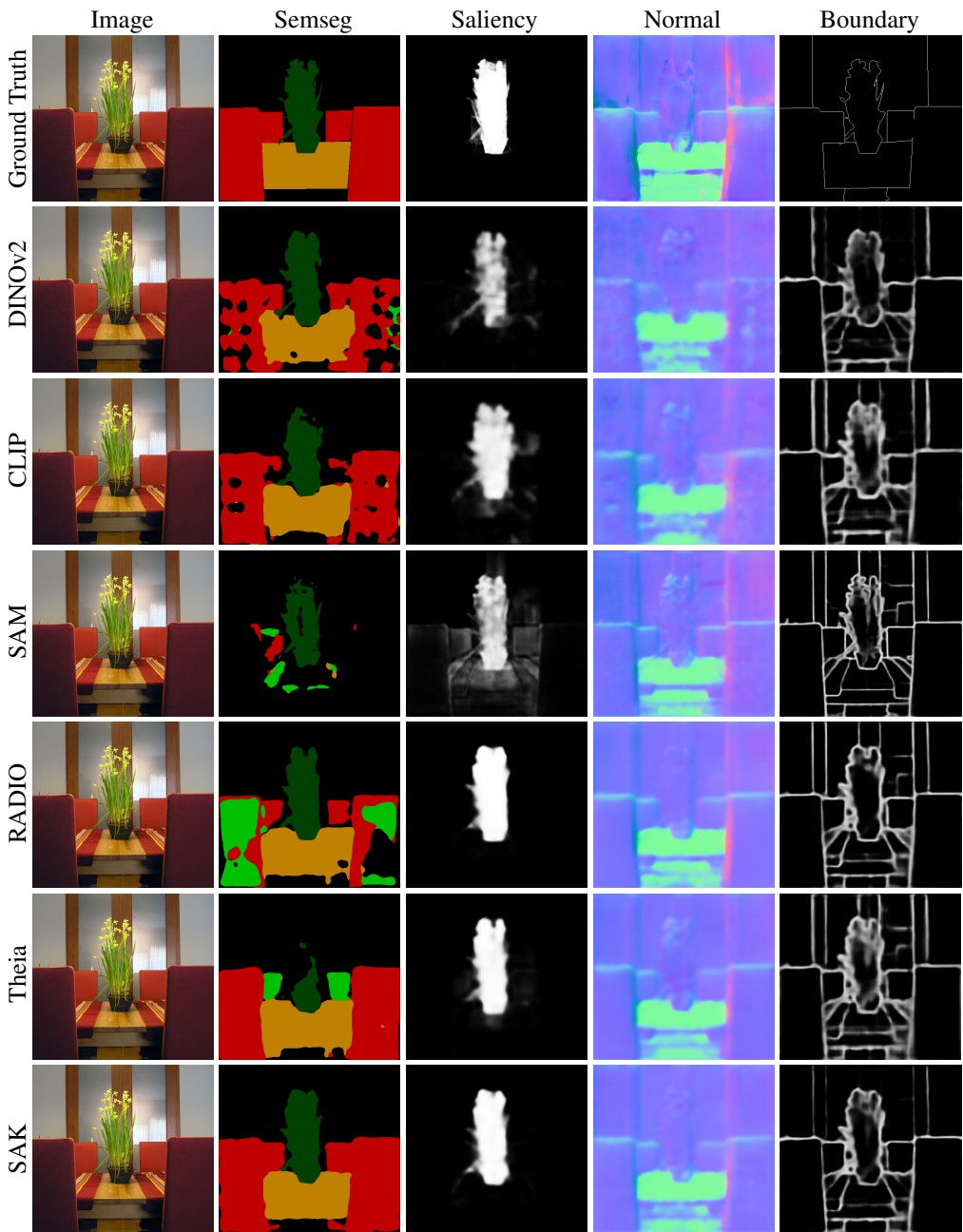

Figure 10: Qualitative results compared with VFM teachers and distillation methods on PASCAL-Context. Note chair and sofa in semantic segmentation.

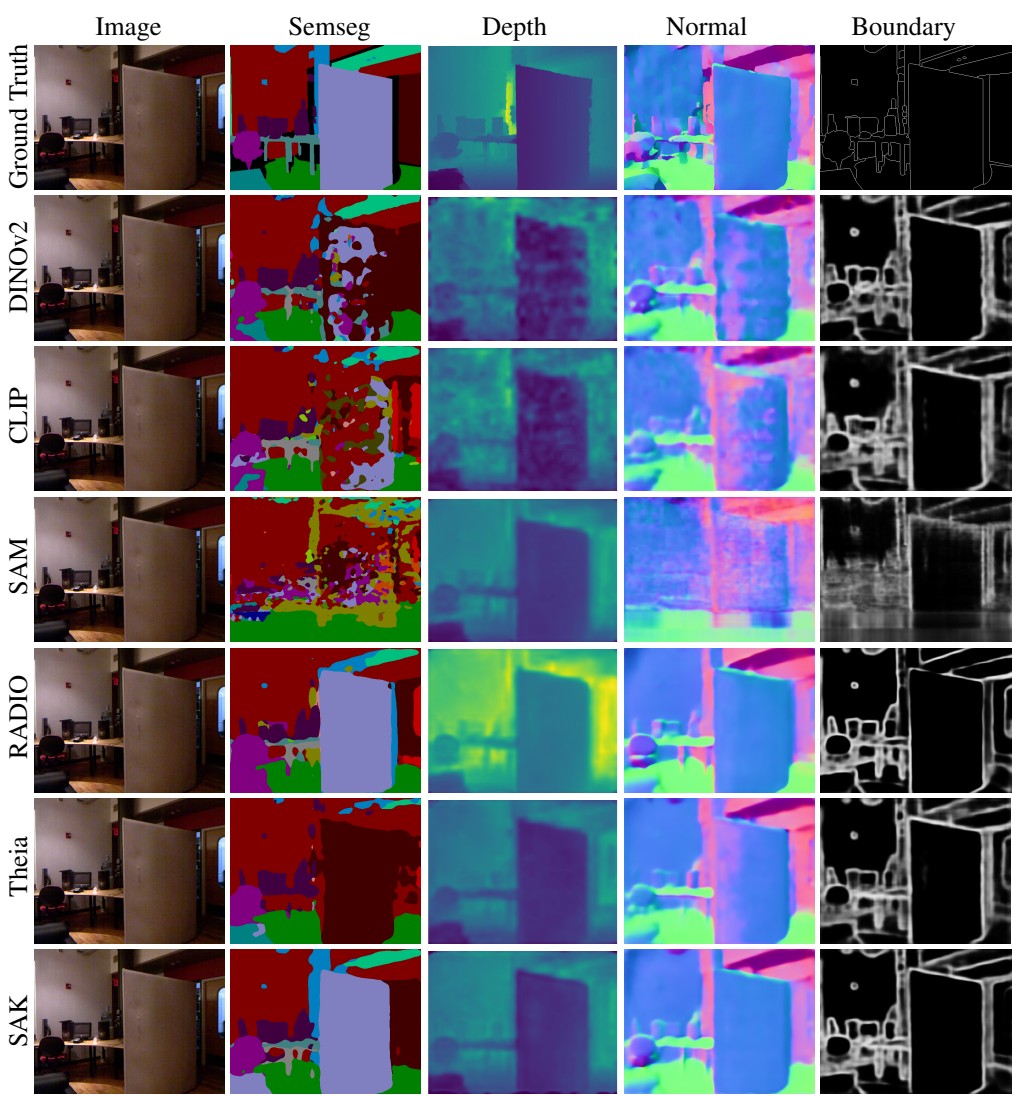

Figure 11: Qualitative results compared with VFM teachers and distillation methods on NYUD-v2.

