# OpenReview forum: "Swiss Army Knife: Synergizing Biases in Knowledge from Vision Foundation Models for Multi-Task Learning"
_ICLR.cc/2025/Conference — ICLR 2025 Poster_

### Official Review · Reviewer_RHQH · 2024-10-27

**Soundness:** 3
**Presentation:** 3
**Contribution:** 3
**Rating:** 8
**Confidence:** 3

**Summary:**

The paper proposes SAK to effectively merge different VFMs for downstream vision tasks, mitigating the issue of the divergence of different biases among the different vision representations generated by each model. SAK innovatively proposes a new paradigm for transferring knowledge with the proposed lightweight Teacher-Specific Adapter Path modules instead of using a standalone backbone in a many-to-one distillation manner. Extensive experiments on two commonly-used benchmarks show the effectiveness of SAK.

**Strengths:**

1. The paper makes a novel attempt to analyze the failures of previous distillation methods. The authors examine the representation biases of VFMs quantitatively and qualitatively to demonstrate that the biases or diversified visual representations have a unique contribution to each downstream task.

2. The proposed adapter path and MoR routers are lightweight and easy to implement, which makes it easy to follow.

3. The experiment results are significant and solid, where the abundant analytical results further prove the effectiveness of the SAK method.

**Weaknesses:**

1. The training difficulty of the router should be discussed since the gating functions of MoE are always hard to optimize for a balanced routing.

2. The authors are encouraged to provide some results on multimodal LLMs (like LLaVA, Sphinx) to further verify the effectiveness of the proposed methods.

**Questions:**

Please refer to the weaknesses section.

---

> ### Author Response · Authors · 2024-11-24
> **Response to Reviewer RHQH (Part 1)**
>
> We appreciate your time and efforts in reviewing our paper, providing valuable comments and constructive suggestions. We are encouraged by your acknowledgement of our work as “a novel attempt to analyze the failures of previous distillation methods,” proposing “lightweight and easy to implement” modules, and demonstrating “significant and solid experiment results.” We provide the following responses to address your concerns.
>
> ### Weakness1
>
> Firstly, our proposed **MoR Router is based on a simple architecture**. As elaborated in Section 3.3 (page 6, line 313), it employs a two-layer MLP design. The noisy gating mechanism, described in Eq. 4, is a standard technique commonly used in MoE approaches [1].
>
> It is important to highlight that, unlike MoE used in previous multi-task learning [2,3,4] or general-purpose models [1,5], where the experts are typically homogeneous during optimization, the experts in our framework **consist of representations distilled from various VFM teachers as well as the Teacher-Agnostic Stem**. These experts **inherently possess different knowledge, making them heterogeneous by design.** As a result, **the MoR Router does not require a balanced routing** because task-specific routers dynamically weigh and combine the most relevant representations for each task. Empirically, **we did not encounter any difficulties in training the MoR Routers.** This approach ensures that the router effectively synergizes the complementary strengths of the diverse representations to optimize performance for different tasks.
>
> We appreciate the reviewer’s detailed feedback and will include this clarification in our manuscript.

---

> ### Author Response · Authors · 2024-11-24
> **Response to Reviewer RHQH (Part 2)**
>
> ### Weakness2
>
> - First, we would like to emphasize that **our work is less targeted at single task learning**, e.g., vision-language learning. Instead, our key motivation lies in the inherent representation biases of VFMs, which result in advantages and disadvantages across different vision tasks. Our SAK synergizes these complementary biases to **enhance multi-task learning**. Additionally, we note that another related baseline, Theia, which focuses on robot learning, does not provide comparisons with RADIO on these tasks.
>
> - Here we supplement the experiments of VLM. Following the setup of RADIO, we integrate SAK into LLaVA-1.5 [6,7], freezing the Teacher-Agnostic Stem and Teacher-Specific Adapter Path, and fine-tuning only the MoR Router. We evaluate three tasks – GQA, POPE, and VQAv2 – at resolutions of 432 and 512, consistent with RADIO. We also include DINOv2-L, CLIP-L, and SAM-H for comparison.
>
> - The following table shows that **while** **SAK is trained on significantly less data, and with smaller VFM teachers, it achieves comparable performance to the teachers and RADIO on GQA and POPE**. The performance gap on VQAv2 is largely attributable to the simplicity of our distillation setup, which prioritizes training efficiency. Regarding dataset scale, as listed in Table 1 (page 2) and Table 20 (Appendix D.3, page 28), **all VFM teachers and RADIO use large-scale datasets**, *e.g.*, DINOv2 uses 142M images, CLIP uses 400M, RADIO uses **1.4B**. In contrast, **our SAK distillation relies on ImageNet-1k**, which has only **1.2M** images. This limited dataset diversity could significantly impact performance on VQA tasks, which require extensive training data.
>
>   Meanwhile, **RADIO employs larger VFM teachers**, including DFN CLIP-H, SigLIP-SO400M, DINOv2-g, and SAM-H, which result in **2.5 times the parameters and forwarding costs** of SAK with teachers based on ViT-L. RADIO also uses a **larger resolution of 1,024 during its distillation** (In LLaVA experiments, the image resolution is 432 or 512), enhancing image information extraction but at a substantial increase in forwarding and backwarding costs. Considering these factors, it is reasonable that SAK cannot outperform RADIO in VLM applications.
>
> | Model          | Resolution | GQA$\uparrow$ | POPE$\uparrow$ | VQAv2$\uparrow$ |
> | :------------- | :--------- | :------------ | :------------- | :-------------- |
> | DINOv2-L/14    | 336        | 62.11         | 87.72          | 76.42           |
> | CLIP-L/14      | 336        | 62.20         | 86.09          | 78.49           |
> | SAM-H/16       | 1024       | 49.92         | 81.76          | 57.65           |
> | RADIOv2.5-B/16 | 432        | 62.09         | 85.87          | 77.24           |
> |                | 512        | 62.70         | 86.59          | 78.03           |
> | RADIOv2.5-L/16 | 432        | 62.89         | 86.13          | 79.44           |
> |                | 512        | 63.58         | 86.66          | 80.04           |
> | SAK-B/16       | 432        | 60.84         | 85.50          | 72.80           |
> |                | 512        | 60.75         | 85.84          | 74.10           |
> | SAK-L/16       | 432        | 62.01         | 86.03          | 75.31           |
> |                | 512        | 62.32         | 86.75          | 75.48           |
>
> - We believe that with access to larger-scale datasets and more powerful teachers during distillation, SAK could achieve competitive or superior results. However, such resources are currently unavailable due to time and computational constraints. We appreciate the reviewer for underscoring the importance of applications in multimodal LLM and will add this discussion and the corresponding results to our manuscript.
>
>
>
> [1] Noam Shazeer, Azalia Mirhoseini, Krzysztof Maziarz, Andy Davis, Quoc Le, Geoffrey Hinton, and Jeff Dean. Outrageously large neural networks: The sparsely-gated mixture-of-experts layer. In ICLR, 2017.
>
> [2] Yuqi Yang, Peng-Tao Jiang, Qibin Hou, Hao Zhang, Jinwei Chen, and Bo Li. Multi-task dense prediction via mixture of low-rank experts. In CVPR, 2024d.
>
> [3] Zitian Chen, Yikang Shen, Mingyu Ding, Zhenfang Chen, Hengshuang Zhao, Erik G LearnedMiller, and Chuang Gan. Mod-Squad: Designing mixtures of experts as modular multi-task learners. In CVPR, 2023.
>
> [4] Hanxue Liang, Zhiwen Fan, Rishov Sarkar, Ziyu Jiang, Tianlong Chen, Kai Zou, Yu Cheng, Cong Hao, and Zhangyang Wang. M3ViT: Mixture-of-experts vision transformer for efficient multitask learning with model-accelerator co-design. In NeurIPS, 2022.
>
> [5] Robert A Jacobs, Michael I Jordan, Steven J Nowlan, and Geoffrey E Hinton. Adaptive mixtures of local experts. Neural computation, 3(1):79–87, 1991.
>
> [6] Haotian Liu, Chunyuan Li, Qingyang Wu, and Yong Jae Lee. Visual instruction tuning. In NeurIPS, 2024.
>
> [7] Haotian Liu, Chunyuan Li, Yuheng Li, and Yong Jae Lee. Improved baselines with visual instruction tuning. In CVPR, 2024.

---

> ### Comment · Reviewer_RHQH · 2024-11-27
>
> The authors address my concerns therefore I raise the score to 8.

---

> > ### Author Response · Authors · 2024-11-27
> >
> > We are glad to know that our response has addressed your concerns. Thank you very much for further acknowledging our paper and raising the score.

---

### Official Review · Reviewer_ok2X · 2024-11-03

**Soundness:** 2
**Presentation:** 3
**Contribution:** 2
**Rating:** 6
**Confidence:** 4

**Summary:**

This work studies the problem of Multi-TaskLearning(MTL) under the context of VisionFoundationModels(VFMs). Specifically, the authors propose the solution termed as “SwissArmyKnife”(SAK), which adaptively distills knowledge from a committee of VFMs to enhance multi-task learning. Experiments on two public benchmarks demonstrates the ability of SAK to synergies the complementary strengths of multiple VFMs.

**Strengths:**

1. Experiments are extensive. The authors have validated their approach on two public datasets across multiple tasks and provide extensive ablation and analysis to demonstrate the effectiveness of the designs.
2. Inference costs is small. It is plausible that the authors try to embed the multiple vision foundation models into a single model which reduces the inference costs of running multiple models significantly.
3. Writing is clear. Overall, the paper is well-organized and the methodology is easy to follow.

**Weaknesses:**

1. Missing comparison with Foundation Models at other tasks. While the authors have compared their method with prior arts and vision foundation models (teachers) on two specific datasets, the reviewer is more interested in how is the performance of the delivered model at the tasks that the vision foundation models are good at? For example, after stage 1 training, how is performance of SAK compared to SAM at semantic segmentation, how is performance of SAK compared to DINOv2 at depth estimation, fine-grained classification, etc. With these study, we could have a more clear picture how well is SAK trained with the help of other vision foundation models.
2. Not always outperforming the best foundation models in Fig. 1. The reviewer noticed from Fig. 1 (left) that SAK cannot outperform the best foundation models at each task (which is sort of expected as they are the teachers), could the authors explain why SAK could outperform the teacher? and why it cannot outperform the teacher at every task?
3. Training costs. Although SAK has reduced the inference costs dramatically, it is expected that it may lead to more training costs as it will iteratively get the prediction from every teacher. The reviewer wonders how is the training and memory costs of SAK compared to baseline methods.

**Questions:**

See weakness above. The reviewer is interested in understanding how well can SAK embed knowledge from multiple vision foundation models and comparisons at the tasks that the vision foundation models are good at will be more convincing.

---

> ### Author Response · Authors · 2024-11-24
> **Response to Reviewer ok2X (Part 1)**
>
> We appreciate your time and efforts in reviewing our paper, providing valuable comments and constructive suggestions. We are encouraged by your acknowledgement of our work in conducting “extensive experiments, ablation and analysis,” ensuring “small inference cost,” and recognizing it as “clear, well-organized” and “easy to follow.” We provide the following responses to address your concerns.
>
> ### Weakness1
>
> We appreciate the reviewer for the insightful question. It is important to note that **performance loss is inevitable during distillation** due to the limited scale of dataset used and the lightweight architecture of SAK. Therefore, we also include a student **distilled from a single teacher** to investigate the help of other teachers during distillation.
>
> - Here we provide an evaluation of the **monocular depth estimation task on the NYUd dataset** [1] (note it is different from the NYUDv2 dataset used in our primary experiments). We follow the evaluation protocol [2] used by DINOv2 and use the identical experimental setups and decode heads (lin. 1 and lin. 4). Since DINOv2 does not provide a complete codebase for this task, we reproduce the pipeline. We also include ViT, SAM, Theia, and baselines reported in the DINOv2 paper, namely OpenCLIP [3], MAE [4], and DINO [5].
>
>   **The effectiveness of synergizing multiple VFM teachers is clearly evidenced by the results**, as **distillation from three teachers improves SAK’s performance** on depth estimation, bringing it closer to the upper bound of the teacher. Notably, it also surpasses other larger foundation models.
>
> | Method                          | Backbone | lin. 1            | lin. 4            |
> | :------------------------------ | :------- | :---------------- | :---------------- |
> |                                 |          | RMSE $\downarrow$ | RMSE $\downarrow$ |
> | ViT                             | ViT-B/16 | 1.118             | 1.117             |
> | SAM                             | ViT-B/16 | 0.678             | 0.652             |
> | Theia                           | ViT-B/16 | 0.644             | 0.629             |
> | OpenCLIP                        | ViT-G/14 | 0.541             | 0.510             |
> | MAE                             | ViT-H/14 | 0.517             | 0.483             |
> | DINO                            | ViT-B/8  | 0.555             | 0.539             |
> | DINOv2 (Teacher upper bound)    | ViT-B/14 | 0.399             | 0.362             |
> | DINOv2 (Our reproduced)         | ViT-B/14 | 0.406             | 0.366             |
> | SAK (Distilled from DINOv2)     | ViT-B/16 | 0.482             | 0.463             |
> | SAK (Distilled from 3 teachers) | ViT-B/16 | 0.450             | 0.436             |
>
> - As mentioned, SAM is specifically designed for promptable segmentation rather than semantic segmentation, which aligns with its semantic-free training objective. Prior studies [6,7] have also demonstrated SAM's limitations in semantic segmentation. In our evaluation, we have assessed SAM’s performance on the benchmarks in Figure 1 (page 1) and Table 12 (Appendix C, page 24), with and without fine-tuning. Similarly, we provide additional results of a student distilled from solely a SAM-B teacher. The results indicate that **SAM struggles to achieve good results with pre-trained knowledge**; while fine-tuning improves its performance, **it still falls short of the performance achieved by SAK**.
>
> | Model                             | PASCAL-Context | NYUDv2         |
> | --------------------------------- | -------------- | -------------- |
> |                                   | mIoU$\uparrow$ | mIoU$\uparrow$ |
> | SAM-B (Freeze)                    | 49.11          | 20.29          |
> | SAM-B (Full Fine-tune)            | 66.39          | 41.92          |
> | SAK-B (Distilled from SAM)        | 63.47          | 43.74          |
> | SAK-B (Distilled from 3 teachers) | 81.65          | 59.02          |

---

> ### Author Response · Authors · 2024-11-24
> **Response to Reviewer ok2X (Part 2)**
>
> ### Weakness2
>
> #### Why cannot SAK outperform teachers at every task?
>
> As noted by the reviewer, **performance loss is inevitable during distillation, since our distillation process is lightweight compared to the extensive pretraining of VFM teachers.**
>
> As detailed in Table 1 (page 3), our VFM teachers are pretrained on large-scale datasets containing hundreds to **thousands of times more image samples** than the ImageNet-1k dataset used for distillation. Moreover, the Teacher-Specific Adapter Path, designed to preserve teachers’ biases, utilizes a **standard and simple architecture** with less than 5% of the backbone’s parameters and computations. Consequently, it is reasonable that the distilled SAK cannot fully inherit teachers’ knowledge when **compressing three teacher models into a single student of similar model size**. Another reason is that SAM operates at a higher input resolution of 1,024, rather than 512 for other models, which bolsters its strength in tasks like boundary detection that rely heavily on fine image details.
>
> #### Why can SAK outperform teachers?
>
> - **The observed superiority of SAK over teachers stems from our motivation to synergize the complementary strengths of teachers.** **Complementarity exists not only across tasks** **but also within tasks**. Take semantic segmentation as an example, as illustrated in Figure 1 and qualitative analysis (Section 2.2, page 4), DINOv2 excels at capturing localized features but CLIP offers strong object-level understanding with its rich semantic knowledge from the language domain. SAM produces exceptional fine-grained pixel-level masks due to its higher resolution. Additional analysis of representations in Appendix A (page 20) further supports these observations. **By preserving the intra-task representation biases during distillation and amalgamating them using proposed Mixture-of-Representations, SAK achieves improved performance.**
>
> - **Another contributing factor is the balance between common knowledge and task-specific information.** Regarding saliency estimation and surface normal estimation, three VFM teachers perform suboptimally when frozen, as they lack downstream task-specific information. Though fine-tuning can alleviate this limitation, it results in degradation of accuracy in segmentation tasks for DINOv2 and CLIP, as shown in the table below (Table 12 in Appendix C, page 24). **This trade-off,** known as the **negative transfer problem** in multi-task learning [8,9]**,** highlights the challenge of balancing task-specific and pretrained knowledge.
>
>   In contrast, SAK preserves the representation biases of the frozen teachers within the TSAP modules while leveraging the Teacher-Agnostic Stem to learn shared knowledge and downstream information. **This disentanglement ensures knowledge diversity without mutual interference.** Moreover, MoR Routers dynamically weigh and combine the most relevant representations for each task, bridging the gap between general-purpose knowledge and task-specific characteristics.
>
> | Model  | Fine-tune | Semseg         | Parsing        | Saliency       | Normal           | Boundary       | $\Delta_m\%$ |
> | :----- | :-------- | :------------- | :------------- | :------------- | :--------------- | :------------- | :----------- |
> |        |           | mIoU$\uparrow$ | mIoU$\uparrow$ | maxF$\uparrow$ | mErr$\downarrow$ | odsF$\uparrow$ | $\uparrow$   |
> | ViT    | MT Full   | 76.76          | 65.26          | 84.39          | 13.98            | 70/37          | \-4.04       |
> | DINOv2 | MT Full   | 77.89          | 70.57          | 84.89          | 13.62            | 74.27          | \-0.56       |
> |        | Freeze    | 81.18          | 74.38          | 80.48          | 16.85            | 70.83          | \-5.39       |
> | CLIP   | MT Full   | 76.84          | 65.87          | 84.61          | 13.91            | 70.50          | \-3.66       |
> |        | Freeze    | 78.33          | 65.31          | 81.48          | 16.84            | 67.43          | \-9.33       |
> | SAM    | MT Full   | 66.39          | 65.65          | 85.38          | 13.74            | 77.20          | \-4.09       |
> |        | Freeze    | 49.11          | 59.85          | 81.10          | 16.21            | 75.89          | \-15.05      |
> | SAK    | MT Full   | 81.65          | 72.38          | 84.87          | 14.05            | 73.23          | \-0.03       |
>
> We appreciate the reviewer for the insightful question and will include this discussion in our manuscript.

---

> ### Author Response · Authors · 2024-11-24
> **Response to Reviewer ok2X (Part 3)**
>
> ### Weakness3
>
> We agree with the reviewer’s point that iteratively forwarding VFM teachers increases training costs, and these costs cannot be overlooked. However, we would like to emphasize that **all multi-teacher distillation methods require** **this process**, and **when compared to other baselines, such as RADIO, SAK demonstrates remarkable efficiency**.
>
> - In the following table, we calculate the number of parameters and computational costs (measured by MACs, Multiply–accumulate operation) for teachers and students in RADIO and SAK, both using a ViT-L backbone for the student. The results show that RADIO incurs **2.5 times the parameters and forwarding costs** of SAK. Additionally, RADIO’s **larger input size of 1,024** introduces more tokens in the representations, further reducing the backward efficiency. Hence, **SAK shows significantly lower demands in RAM, GPU memory, training time, and storage.**
>
> | **RADIO teachers and students**    | Input size | #Param$\downarrow$ | MACs$\downarrow$ | **SAK teachers and students**      | Input size | #Param$\downarrow$ | MACs$\downarrow$ |
> | :--------------------------------- | :--------- | :----------------- | :--------------- | :--------------------------------- | :--------- | :----------------- | :--------------- |
> | DFN CLIP-H/14                      | 378        | 631M               | 460G             | DINOv2-L/14                        | 384        | 304M               | 221G             |
> | SigLIP-SO400M/14                   | 384        | 413M               | 300G             | OpenCLIP-L/14                      | 384        | 304M               | 221G             |
> | DINOv2-g/14                        | 224        | 1135M              | 291G             | SAM-L/16                           | 1024       | 307M               | 1308G            |
> | SAM-H/16                           | 1024       | 636M               | 2730G            |                                    |            |                    |                  |
> | RADIOv2.5-L/16                     | 1024       | 320M               | 1240G            | SAK-L/16                           | 384        | 343M               | 198G             |
> | Sum over all teachers and students |            | **3135M**          | **5021G**        | Sum over all teachers and students |            | **1258M**          | **1948G**        |
>
> - Moreover, as listed in Table 20 (Appendix D.3, page 28), RADIO is trained on the large-scale DataComp-1B dataset (**1.4B images**) with **614M total samples** seen. In contrast, SAK is trained on ImageNet-1k dataset (**1.2M images**) with **only 36M samples** seen, further highlighting the efficiency of SAK in training and memory costs.
>
> We appreciate the reviewer for highlighting the importance of training costs. We will include this analysis and discussion in our manuscript.
>
>
>
> [1] Nathan Silberman, Derek Hoiem, Pushmeet Kohli, and Rob Fergus. Indoor segmentation and support inference from RGBD images. In ECCV, 2012.
>
> [2] Zhenyu Li, Xuyang Wang, Xianming Liu, and Junjun Jiang. Binsformer: Revisiting adaptive bins for monocular depth estimation. IEEE Transactions on Image Processing, 2024.
>
> [3] Mehdi Cherti, Romain Beaumont, Ross Wightman, Mitchell Wortsman, Gabriel Ilharco, Cade Gordon, Christoph Schuhmann, Ludwig Schmidt, and Jenia Jitsev. Reproducible scaling laws for contrastive language-image learning. In CVPR, 2023.
>
> [4] Kaiming He, Xinlei Chen, Saining Xie, Yanghao Li, Piotr Dollár, and Ross Girshick. Masked autoencoders are scalable vision learners. In CVPR, 2022.
>
> [5] Mathilde Caron, Hugo Touvron, Ishan Misra, Hervé Jégou, Julien Mairal, Piotr Bojanowski, and Armand Joulin. Emerging properties in self-supervised vision transformers. In CVPR, 2021.
>
> [6] Mike Ranzinger, Greg Heinrich, Jan Kautz, and Pavlo Molchanov. AM-RADIO: Agglomerative vision foundation model reduce all domains into one. In CVPR, 2024.
>
> [7] Haoxiang Wang, Pavan Kumar Anasosalu Vasu, Fartash Faghri, Raviteja Vemulapalli, Mehrdad Farajtabar, Sachin Mehta, Mohammad Rastegari, Oncel Tuzel, and Hadi Pouransari. SAM-CLIP: Merging vision foundation models towards semantic and spatial understanding. In CVPR, 2024a.
>
> [8] Simon Vandenhende, Stamatios Georgoulis, Wouter Van Gansbeke, Marc Proesmans, Dengxin Dai, and Luc Van Gool. Multi-task learning for dense prediction tasks: A survey. TPAMI, 44(7): 3614–3633, 2021.
>
> [9] Simon Vandenhende, Stamatios Georgoulis, and Luc Van Gool. MTI-Net: Multi-scale task interaction networks for multi-task learning. In ECCV, 2020.

---

> > ### Comment · Reviewer_ok2X · 2024-11-24
> > **reply**
> >
> > Thanks for the detailed reply by the authors.
> >
> > It has addressed most of my concerns and I will increase my rating to 6 accordingly.

---

> > > ### Author Response · Authors · 2024-11-25
> > > **Thank you!**
> > >
> > > We are glad that our response has addressed your concerns. Thank you very much for acknowledging our paper and updating the rating.

---

### Official Review · Reviewer_yJkf · 2024-11-03

**Soundness:** 2
**Presentation:** 2
**Contribution:** 2
**Rating:** 6
**Confidence:** 3

**Summary:**

This paper introduces a novel method to consolidate different visual foundation models into one model via a mixture-of-expert fashion method. Through their approach, their method Swiss Army Knife (SAK) achieve state-of-the-art performance across various vision tasks.

**Strengths:**

(1) This paper systematically study the representation bias in different visual foundation models (VFMs), and identify characteristics of VFMs.
(2) The proposed approach achieve state-of-the-art performance on multi-task learning on various vision tasks.
(3) Thorough comparisons with baseline methods are provided to validate the effectiveness of the method. In addition, visualizations on choices of experts are given to provide insights into the method.

**Weaknesses:**

(1) The proposed mixture-of-representation router involves conducting weighted sum over both student's and teacher's features, which would increase the inference cost during the process. This is because all teacher models will have to conduct forward propagation to obtain the output representations.
(2) More vision tasks such has instance-level segmentation/detection, depth estimation, could be evaluated.

**Questions:**

One thing the reviewer tries to seek clarification is how is mixture-of-representation being done when student and teacher has different architecture. In this case, the feature dimension of student and teacher is different.

---

> ### Author Response · Authors · 2024-11-24
> **Response to Reviewer yJkf (Part 1)**
>
> We appreciate your time and efforts in reviewing our paper, providing valuable comments and constructive suggestions. We are encouraged by your acknowledgement of our work as a “novel method” that achieves “state-of-the-art performance” based on “thorough comparisons,” and offers “insights.” We provide the following responses to address your concerns.
>
> ### Weakness1
>
> We would like to clarify a potential misunderstanding of the inference procedure. **Our framework does not require teachers during inference**. Instead, the representations distilled from the teachers are generated directly by the lightweight TSAP modules, **which introduce minimal additional parameters and computational overhead.**
>
> To provide a clearer perspective, we present the number of parameters and computational cost (measured by MACs, Multiply–accumulate operations) introduced by our modules when integrated into the ViT-B or ViT-L backbone. The calculation is based on a student model corresponding to three teachers on the five-task PASCAL-Context dataset. The table indicates that **each TSAP branch accounts for less than 5% of the backbone’s parameters and computations** (4M *vs.* 86M parameters, 4G *vs.* 88G MACs with ViT-B). Similarly, **each MoR Router is as lightweight as 2M parameters and 2G MACs**, even with the larger ViT-L backbone.
>
> | Model               | \#Param$\downarrow$ | MACs$\downarrow$ | Model               | \#Param$\downarrow$ | MACs$\downarrow$ |
> | :------------------ | :------------------ | :--------------- | :------------------ | :------------------ | :--------------- |
> | ViT-B/16            | 86M                 | 88G              | ViT-L/16            | 304M                | 311G             |
> | \+TSAP (3 branches) | 98M                 | 100G             | \+TSAP (3 branches) | 344M                | 351G             |
> | \+MoR (5 tasks)     | 104M                | 106G             | \+MoR (5 tasks)     | 354M                | 362G             |
>
> We appreciate the reviewer for highlighting this concern and will add related discussion on inference cost to our manuscript.
>
> ### Weakness2
>
> - We would like to highlight that **our work targets multi-task learning**, and **the tasks we evaluated are defined by the dataset**, since each image sample must be labeled for every task included. PASCAL-Context and NYUDv2 are multi-task benchmarks used by most recent works [1,2,3], allowing direct comparisons with state-of-the-art methods. We would also like to note that **depth estimation is already included in the NYUDv2 dataset**.
>
>
> - Due to time constraints, we currently supplement an evaluation of the **monocular depth estimation task on the NYUd dataset** [4] (note it is different from the NYUDv2 dataset). We follow the evaluation protocol [5] used by DINOv2 and use the identical experimental setups and decode heads (lin. 1 and lin. 4). Since DINOv2 does not provide a complete codebase for this task, we reproduce the pipeline. We also include ViT, SAM, Theia, and baselines reported in the DINOv2 paper, namely OpenCLIP [6], MAE [7], and DINO [8].
>
>   **While SAK does not fully match the teacher’s performance due to inevitable loss during distillation and different patch size, it surpasses other foundation models and baselines**, even those with larger backbones. As noted in Table 1 (page 3), DINOv2 is trained on the **large-scale LVD-142M dataset,** which contains **over 100 times more image samples** than the ImageNet-1k dataset we use for distillation (CLIP and SAM are trained on even huger datasets). Hence, **it is reasonable that our approach cannot outperform the teachers on their proficient tasks**. We believe that leveraging larger datasets could further enhance knowledge transfer during distillation and benefit downstream tasks. We will include this analysis in our manuscript.
>
> | Method                       | Backbone | lin. 1            | lin. 4            |
> | ---------------------------- | -------- | ----------------- | ----------------- |
> |                              |          | RMSE $\downarrow$ | RMSE $\downarrow$ |
> | ViT                          | ViT-B/16 | 1.118             | 1.117             |
> | SAM                          | ViT-B/16 | 0.678             | 0.652             |
> | Theia                        | ViT-B/16 | 0.644             | 0.629             |
> | OpenCLIP                     | ViT-G/14 | 0.541             | 0.510             |
> | MAE                          | ViT-H/14 | 0.517             | 0.483             |
> | DINO                         | ViT-B/8  | 0.555             | 0.539             |
> | DINOv2 (Teacher upper bound) | ViT-B/14 | 0.399             | 0.362             |
> | DINOv2 (Our reproduced)      | ViT-B/14 | 0.406             | 0.366             |
> | SAK (Ours)                   | ViT-B/16 | 0.450             | 0.436             |

---

> ### Author Response · Authors · 2024-11-24
> **Response to Reviewer yJkf (Part 2)**
>
> ### Question1
>
> We appreciate the reviewer’s detailed feedback. **We have already illustrated the teacher/student mismatch issue in Appendix B.1 (page 21, line 1127).** When the student and teacher have different architectures, discrepancies could arise in the spatial resolutions due to varying input sizes or patch sizes and in the number of channels due to diverse embedding dimensions. In the Teacher-Specific Adapter Paths, all representations maintain the same shape as those in Teacher-Agnostic Stem. To align the student’s representations with those of the teachers, **we introduce an upsampling layer to match the spatial resolution, and a linear layer for channel mapping when necessary**, following established practices [9,10]. Note that the linear layer is only required during distillation and does not add any overhead during inference.
>
>
>
> [1] Hanrong Ye and Dan Xu. TaskPrompter: Spatial-channel multi-task prompting for dense scene understanding. In ICLR, 2023b.
>
> [2] Hanrong Ye and Dan Xu. TaskExpert: Dynamically assembling multi-task representations with memorial mixture-of-experts. In ICCV, 2023a.
>
> [3] Yuqi Yang, Peng-Tao Jiang, Qibin Hou, Hao Zhang, Jinwei Chen, and Bo Li. Multi-task dense prediction via mixture of low-rank experts. In CVPR, 2024d.
>
> [4] Nathan Silberman, Derek Hoiem, Pushmeet Kohli, and Rob Fergus. Indoor segmentation and support inference from RGBD images. In ECCV, 2012.
>
> [5] Zhenyu Li, Xuyang Wang, Xianming Liu, and Junjun Jiang. Binsformer: Revisiting adaptive bins for monocular depth estimation. IEEE Transactions on Image Processing, 2024.
>
> [6] Mehdi Cherti, Romain Beaumont, Ross Wightman, Mitchell Wortsman, Gabriel Ilharco, Cade Gordon, Christoph Schuhmann, Ludwig Schmidt, and Jenia Jitsev. Reproducible scaling laws for contrastive language-image learning. In CVPR, 2023.
>
> [7] Kaiming He, Xinlei Chen, Saining Xie, Yanghao Li, Piotr Dollár, and Ross Girshick. Masked autoencoders are scalable vision learners. In CVPR, 2022.
>
> [8] Caron, Mathilde, Hugo Touvron, Ishan Misra, Hervé Jégou, Julien Mairal, Piotr Bojanowski, and Armand Joulin. Emerging properties in self-supervised vision transformers. In CVPR, 2021.
>
> [9] Mike Ranzinger, Greg Heinrich, Jan Kautz, and Pavlo Molchanov. AM-RADIO: Agglomerative vision foundation model reduce all domains into one. In CVPR, 2024.
>
> [10] Shengcao Cao, Mengtian Li, James Hays, Deva Ramanan, Yu-Xiong Wang, and Liang-Yan Gui. Learning lightweight object detectors via multi-teacher progressive distillation. In ICML, 2023.

---

> ### Author Response · Authors · 2024-12-01
>
> Dear Reviewer yJkf,
>
> We sincerely appreciate the time and effort you have dedicated to reviewing our submission. Thank you once again for your valuable and constructive comments, and we have incorporated the results of our discussion into the revised manuscript.
>
> Given that the last day that reviewers can post responses is December 2, we would like to know if you have any additional comments. We would be more than happy to include them before the discussion phase ends.
>
> Best regards,
>
> Authors

---

### Official Review · Reviewer_gxTF · 2024-11-04

**Soundness:** 3
**Presentation:** 3
**Contribution:** 2
**Rating:** 6
**Confidence:** 3

**Summary:**

This paper proposes a new method to achieve multi-task learning by coordinating the advantages of multiple Vision Foundation Models (VFMs). In order to solve the problem that the previous distillation method only uses a single student model that cannot preserve the representation bias between different VFMs, a teacher-agnostic stem and a teacher-specific adapter path modules are proposed to parameter-efficiently preserve the representation bias of different VFMs, and then the fusion coefficients of different branches are learned for different tasks through a mixture-of-representations router. Experiments on the PASCAL-Context and NYUD-v2 datasets have demonstrated the effectiveness of the method.

**Strengths:**

1. By distilling multiple Vision Foundation Models (VFMs) into a stem and multiple parameter-efficient branches, the computational cost is greatly reduced

2. By retaining the unique representation bias of each teacher model through the teacher-specific adapter path module, SAK can extract knowledge from multiple VFMs in a task-adaptive manner, thereby effectively improving multi-task performance

3. Comparative experiments and sufficient ablation experiments on PASCAL-Context and NYUD-v2 datasets demonstrate the effectiveness of the proposed method

**Weaknesses:**

1. Lack of experimental results on Image level reasoning and Large Vision-Language Model that are consistent with the comparison method RADIO

2. Lack of upper bounds on results before distillation, i.e., results using encoders of three Vision Foundation Models without distillation and routers

**Questions:**

1. Compared with the base model selected in this paper, RADIO also selected SigLIP and Theia selected ViT-H. It is necessary to provide the principles of base model selection and experiments on the sensitivity of the model to teacher selection.

2. Can SAK be easily adjusted to add new Vision Foundation Models teachers or changes that meet the needs of specific downstream tasks?

---

> ### Author Response · Authors · 2024-11-24
> **Response to Reviewer gxTF (Part 1)**
>
> We appreciate your time and efforts in reviewing our paper, providing valuable comments and constructive suggestions. We are encouraged by your acknowledgement of our work in achieving “reduced computational cost” while “effectively improving multi-task performance,” as demonstrated by the “sufficient ablation experiments.” We provide the following responses to address your concerns.
>
> ### Weakness1
>
> - First, we would like to emphasize that **our work is less targeted at single task learning**, e.g., image level reasoning or vision-language learning. Instead, our key motivation lies in the inherent representation biases of VFMs, which result in advantages and disadvantages across different vision tasks. Our SAK synergizes these complementary biases to **enhance multi-task learning**. Additionally, we note that another related baseline, Theia, which focuses on robot learning, does not provide comparisons with RADIO on these tasks.
>
>
> - Here we supplement the experiments of VLM. Following the setup of RADIO, we integrate SAK into LLaVA-1.5 [1,2], freezing the Teacher-Agnostic Stem and Teacher-Specific Adapter Path, and fine-tuning only the MoR Router. We evaluate three tasks – GQA, POPE, and VQAv2 – at resolutions of 432 and 512, consistent with RADIO. We also include DINOv2-L, CLIP-L, and SAM-H for comparison.
>
> - The following table shows that **while** **SAK is trained on significantly less data, and with smaller VFM teachers, it achieves comparable performance to the teachers and RADIO on GQA and POPE**. The performance gap on VQAv2 is largely attributable to the simplicity of our distillation setup, which prioritizes training efficiency. Regarding dataset scale, as listed in Table 1 (page 2) and Table 20 (Appendix D.3, page 28), **all VFM teachers and RADIO use large-scale datasets**, *e.g.*, DINOv2 uses 142M images, CLIP uses 400M, RADIO uses **1.4B**. In contrast, **our SAK distillation relies on ImageNet-1k**, which has only **1.2M** images. This limited dataset diversity could significantly impact performance on VQA tasks, which require extensive training data.
>
>   Meanwhile, **RADIO employs larger VFM teachers**, including DFN CLIP-H, SigLIP-SO400M, DINOv2-g, and SAM-H, which result in **2.5 times the parameters and forwarding costs** of SAK with teachers based on ViT-L. RADIO also uses a **larger resolution of 1,024 during its distillation** (In LLaVA experiments, the image resolution is 432 or 512), enhancing image information extraction but at a substantial increase in forwarding and backwarding costs. Considering these factors, it is reasonable that SAK cannot outperform RADIO in VLM applications.
>
> | Model          | Resolution | GQA$\uparrow$ | POPE$\uparrow$ | VQAv2$\uparrow$ |
> | :------------- | :--------- | :------------ | :------------- | :-------------- |
> | DINOv2-L/14    | 336        | 62.11         | 87.72          | 76.42           |
> | CLIP-L/14      | 336        | 62.20         | 86.09          | 78.49           |
> | SAM-H/16       | 1024       | 49.92         | 81.76          | 57.65           |
> | RADIOv2.5-B/16 | 432        | 62.09         | 85.87          | 77.24           |
> |                | 512        | 62.70         | 86.59          | 78.03           |
> | RADIOv2.5-L/16 | 432        | 62.89         | 86.13          | 79.44           |
> |                | 512        | 63.58         | 86.66          | 80.04           |
> | SAK-B/16       | 432        | 60.84         | 85.50          | 72.80           |
> |                | 512        | 60.75         | 85.84          | 74.10           |
> | SAK-L/16       | 432        | 62.01         | 86.03          | 75.31           |
> |                | 512        | 62.32         | 86.75          | 75.48           |
>
> - We believe that with access to larger-scale datasets and more powerful teachers during distillation, SAK could achieve competitive or superior results. However, such resources are currently unavailable due to time and computational constraints. We will add this discussion and the corresponding results to our manuscript.

---

> ### Author Response · Authors · 2024-11-24
> **Response to Reviewer gxTF (Part 2)**
>
> ### Weakness2
>
> - We build an upper-bound model using the frozen encoders of three VFM teachers – DINOv2, CLIP, and SAM – along with a learnable ViT encoder as a surrogate for the Teacher-Agnostic Stem (TAS). All components employ a ViT-B backbone. For fusing the representations, we consider element-wise addition, channel concatenation, and our proposed Mixture-of-Representations.
>
>
> - The following table shows that a naive addition of representations fails to fully leverage the teachers’ knowledge, as mentioned in the introduction (page 2, line 84) and further supported by our ablation study in Section 4.3 (page 8, line 424) and Table 3 (page 8). While concatenation addresses this limitation and leads to upper-bound performance, **it significantly increases the parameter count and computational overhead** since it increases the dimension of the fused representations. In contrast, **our MoR approach surpasses addition by 0.5% while maintaining comparable parameters and MACs**, demonstrating its effectiveness even in this alternative setup.
> - Compared to these upper bounds, **our SAK outperforms the naive addition with around 1/3 of the parameters and half of the computational cost.** This further validates the effectiveness and efficiency of our method. We appreciate the reviewer’s valuable suggestion and will add the upper bound results into our manuscript.
>
> |             | Fuse     | \#Params     | MACs         | Semseg         | Parsing        | Saliency       | Normal           | Boundary       | $\Delta_m\%$ |
> | :---------- | :------- | :----------- | :----------- | :------------- | :------------- | :------------- | :--------------- | :------------- | :----------- |
> |             |          | $\downarrow$ | $\downarrow$ | mIoU$\uparrow$ | mIoU$\uparrow$ | maxF$\uparrow$ | mErr$\downarrow$ | odsF$\uparrow$ | $\uparrow$   |
> | Upper bound | Addition | 378M         | 1091G        | 80.27          | 71.44          | 84.74          | 13.82            | 72.69          | \-0.47       |
> |             | Concat   | 820M         | 7661G        | 82.26          | 73.85          | 84.63          | 13.98            | 75.55          | 1.21         |
> |             | MoR      | 384M         | 1097G        | 80.58          | 72.73          | 84.88          | 14.05            | 74.01          | 0.02         |
> | SAK         | MoR      | 134M         | 544G         | 81.65          | 72.38          | 84.87          | 14.05            | 73.23          | \-0.03       |

---

> ### Author Response · Authors · 2024-11-24
> **Response to Reviewer gxTF (Part 3)**
>
> ### Question1
>
> #### Principles of VFM teacher selection
>
> - It is well-established that **DINOv2, CLIP, and SAM are among the most widely used Vision Foundation Models.** Prior baselines RADIO and Theia also employ them for experiments. Besides, RADIO uses DFN CLIP and SigLIP, both of which are improved models derived from CLIP. However, as shown in the following table from our early attempts, **DFN CLIP and SigLIP fail to attain comparable outcomes to CLIP**. For this reason, we choose to adhere to CLIP when selecting teachers.
>
> | Model    | Fine-tune | Semseg         | Parsing        | Saliency       | Normal           | Boundary       | $\Delta_m\%$ |
> | :------- | :-------- | :------------- | :------------- | :------------- | :--------------- | :------------- | :----------- |
> |          |           | mIoU$\uparrow$ | mIoU$\uparrow$ | maxF$\uparrow$ | mErr$\downarrow$ | odsF$\uparrow$ | $\uparrow$   |
> | CLIP     | MT Full   | 76.84          | 65.87          | 84.61          | 13.91            | 70.50          | \-3.66%      |
> |          | Freeze    | 78.33          | 65.31          | 81.48          | 16.84            | 67.43          | \-9.33%      |
> | DFN CLIP | MT Full   | 75.19          | 63.78          | 84.63          | 14.00            | 70.57          | \-4.77%      |
> |          | Freeze    | 69.35          | 53.48          | 81.36          | 17.24            | 64.85          | \-16.24%     |
> | SigLIP   | MT Full   | 75.32          | 63.57          | 84.67          | 14.00            | 70.50          | \-4.81%      |
> |          | Freeze    | 51.72          | 45.34          | 77.93          | 18.35            | 60.33          | \-26.61%     |
>
> - Theia utilizes ViT (pretrained on ImageNet) and Depth Anything [3] as additional teachers. Since the **Teacher-Agnostic Stem in SAK is initialized with an ImageNet-pretrained ViT backbone, we do not include ViT as an additional teacher to reduce training costs**. While Depth Anything is a high-impact VFM, we leave its exploration for future work due to limited time and resources.
>
>
> - Additionally, we **avoid using very large models** (e.g. ViT-H or ViT-g based) in our experiments due to constraints in computational resources (RADIO uses 64 GPUs, while we can only use 8 GPUs). We would also like to mention that SAK is a highly flexible framework, capable of integrating and benefiting further from more powerful VFM teachers, making it adaptable to various settings.
>
> #### Sensitivity of SAK to VFM teacher selection
>
> - We have already provided a detailed **analysis of teacher selection in Section 4.3** (page 9, line 459). Table 7 (page 9) shows the performance experimented using different combinations of teachers. The results indicate that **integrating knowledge from all three teachers leads to the strongest overall performance**. Table 8 (page 10) and its analysis (page 9, line 476) further show that **scaling up the teacher models proves beneficial**.
>
>
> - Here we supplement another analysis with a SAK-B student distilled from DINOv2-B and SigLIP-B teachers. Compared with the student distilled from DINOv2-B and CLIP-B teachers, it exhibits better performance on three out of five tasks, while achieving comparable overall accuracy, further **underscoring the robustness and adaptability of our method**. We will include the above discussions in our manuscript.
>
> | Teachers           | Semseg         | Parsing        | Saliency       | Normal           | Boundary       | $\Delta_m\%$ |
> | :----------------- | :------------- | :------------- | :------------- | :--------------- | :------------- | :----------- |
> |                    | mIoU$\uparrow$ | mIoU$\uparrow$ | maxF$\uparrow$ | mErr$\downarrow$ | odsF$\uparrow$ | $\uparrow$   |
> | DINOv2-B, CLIP-B   | **81.53**      | **71.95**      | 84.49          | 14.16            | 72.59          | **-0.60**    |
> | DINOv2-B, SigLIP-B | 81.41          | 71.03          | **84.62**      | **14.06**        | **72.92**      | \-0.63       |

---

> ### Author Response · Authors · 2024-11-24
> **Response to Reviewer gxTF (Part 4)**
>
> ### Question2
>
> Yes, SAK is a highly flexible framework that can be **easily adjusted** to add new teachers or new downstream tasks.
>
> #### Add new VFM teachers
>
> When adding a new teacher, we freeze the existing Teacher-Agnostic Stem and Teacher-Specific Adapter Paths to preserve the knowledge transferred from the previous teachers A new TSAP module is then introduced and distilled for the new teacher, **facilitating efficient adaptation**. In the second stage, we train on the downstream dataset with all VFM teachers and tune the entire student model.
>
> We conduct an experiment by incorporating a new SigLIP teacher into a student already distilled from DINOv2, CLIP, and SAM. As shown in the following table, **including SigLIP teacher boosts performance on Semseg and Normal tasks and yields competitive results on Saliency and Boundary tasks**. However, since SigLIP encodes knowledge similar to CLIP during pretraining and does not surpass CLIP on the benchmark, it is reasonable that it could not lead to further improvements. Moreover, the additional representations may increase the difficulty for the MoR Routers in learning optimal weights, potentially accounting for the degradation in human parsing.
>
> | Teachers                    | Semseg         | Parsing        | Saliency       | Normal           | Boundary       | $\Delta_m\%$ |
> | :-------------------------- | :------------- | :------------- | :------------- | :--------------- | :------------- | :----------- |
> |                             | mIoU$\uparrow$ | mIoU$\uparrow$ | maxF$\uparrow$ | mErr$\downarrow$ | odsF$\uparrow$ | $\uparrow$   |
> | DINOv2-B, OpenCLIP-B, SAM-B | 81.65          | **72.38**      | **84.87**      | 14.05            | **73.23**      | **-0.03**    |
> | \+SigLIP-B                  | **81.78**      | 70.66          | 84.85          | **14.02**        | 73.21          | \-0.45       |
>
> #### Add different tasks
>
> Additionally, when applying to different downstream tasks, **it is only necessary to add task-specific decoders and perform the second-stage training** (with distillation) or **fine-tuning** (without distillation), as the first-stage distillation is agnostic to the downstream tasks. This process is significantly more efficient than the first stage. Meanwhile, **we can readily remove a branch of TSAP** if it is no longer necessary**,** without affecting the functionality of the TAS or other TSAP branches, ensuring the **modularity and adaptability** of our design. We have also shown in Figure 5 (page 10) and its accompanying analysis (page 9, line 470) that SAK **exhibits strong robustness and generalization capability in downstream tasks**.
>
> We appreciate the reviewer for the insightful question and will add this discussion and experimental results to our manuscript.
>
>
>
> [1] Haotian Liu, Chunyuan Li, Qingyang Wu, and Yong Jae Lee. Visual instruction tuning. In NeurIPS, 2024.
>
> [2] Haotian Liu, Chunyuan Li, Yuheng Li, and Yong Jae Lee. Improved baselines with visual instruction tuning. In CVPR, 2024.
>
> [3] Lihe Yang, Bingyi Kang, Zilong Huang, Xiaogang Xu, Jiashi Feng, and Hengshuang Zhao. Depth Anything: Unleashing the power of large-scale unlabeled data. In CVPR, 2024b.

---

> ### Author Response · Authors · 2024-11-27
> **Response to Reviewer gxTF (Part 5)**
>
> We would like to provide updates based on additional experiments on image level reasoning. Following the setup of DINOv2, we evaluate linear probing on the ImageNet-1k dataset. We freeze the backbones and train a linear classifier and the MoR Routers of SAK using the same hyperparameters as DINOv2. For comparison, we also evaluate the VFM teachers and the state-of-the-art baseline RADIO, while reporting Theia’s result from its original paper.
>
> The following table shows that **SAK outperforms the two baselines**, despite a performance gap compared to the DINOv2 and CLIP teachers due to inevitable loss during distillation. It is worth noting that RADIO uses more powerful teachers, including DINOv2-g/14-reg (accuracy 87.1), while Theia uses DINOv2-L/14 teacher (accuracy 86.3).
>
> | Model          | Accuracy$\uparrow$ |
> | -------------- | ------------------ |
> | DINOv2-B/14    | 84.5               |
> | CLIP-B/16      | 84.7               |
> | SAM-B/16       | 46.9               |
> | RADIOv2.5-B/16 | 78.2               |
> | Theia-B/16     | 75.2               |
> | SAK-B/16       | 79.1               |
>
> Please feel free to let us know if you have any additional comments, we would be happy to discuss them.

---

> ### Author Response · Authors · 2024-12-01
>
> Dear Reviewer gxTF,
>
> We sincerely appreciate the time and effort you have dedicated to reviewing our submission. Thank you once again for your valuable and constructive comments, and we have incorporated the results of our discussion into the revised manuscript.
>
> Given that the last day that reviewers can post responses is December 2, we would like to know if you have any additional comments. We would be more than happy to include them before the discussion phase ends.
>
> Best regards,
>
> Authors

---

### Official Review · Reviewer_tiXZ · 2024-11-04

**Soundness:** 2
**Presentation:** 3
**Contribution:** 2
**Rating:** 6
**Confidence:** 3

**Summary:**

This paper introduces a "Swiss Army Knife" (SAK) model that preserves each model's specific strengths through a framework consisting of a shared Teacher-Agnostic Stem and Teacher-Specific Adapter Paths. The SAK dynamically combines representations via Mixture-of-Representations Routers, allowing for tailored outputs for each task. Extensive experiments on multi-task benchmarks show that SAK outperforms several current methods.

**Strengths:**

1. This paper contributes a multi-task learning framework that leverages the strengths and specific biases of multiple Vision Foundation Models (VFMs), presenting an alternative to traditional knowledge distillation approaches. By preserving individual model biases, the method offers a good solution to challenges in multi-task learning, where distinct vision tasks often benefit from different aspects of visual representation.

2. Extensive experimental results are provided, showing substantial performance gains across established benchmarks.

3. This paper is well organized and well written.

**Weaknesses:**

1. The proposed framework, while innovative, introduces a high level of algorithmic complexity by requiring multiple Vision Foundation Models (VFMs) and integrating Teacher-Specific Adapter Paths and Mixture-of-Representations Routers. Given the inherent complexity of multi-task learning, this layered structure may lead to excessive computational overhead without clear evidence of structural necessity. The paper could improve by providing a more rigorous theoretical justification for this architecture, perhaps through ablation studies that explore simpler configurations to assess if comparable results could be achieved with fewer components.

2. The impressive experimental results may be partly due to extensive parameter tuning, but the paper lacks detailed discussions or tests that could isolate the contributions of the model’s design from effects arising purely from tuning. This raises questions about whether the performance gains reflect true architectural benefits or if they could be replicated by tuning existing simpler models. Including experiments with fixed hyperparameters across different tasks or using cross-validation techniques to verify robustness would strengthen confidence in the model’s structural contributions.

**Questions:**

please see the weaknesses section.

---

> ### Author Response · Authors · 2024-11-24
> **Response to Reviewer tiXZ (Part 1)**
>
> We appreciate your time and efforts in reviewing our paper, providing valuable comments and constructive suggestions. We are encouraged by your acknowledgement of our work as “a good solution” in multi-task learning, noting its “substantial performance gains” and recognizing it as a “well organized and well written” paper. We provide the following responses to address your concerns.
>
> ### Weakness1
>
> We would like to emphasize that **our module architectures are simple,** introducing **minimal additional parameters and computational overhead** while achieving **significant improvements** over the previous state of the art, which also requires multiple VFMs. Furthermore, compared to prior multi-task approaches that rely on complicated module designs [1,2,3], our architecture is **much simpler, with fewer parameters**, yet delivers **substantial performance gains**, as shown in Tables 5 (page 8) and 6 (page 9).
>
> - As described in Section 3.2 (page 6, line 283), the adapter follows a **standard structure** consisting of a **two-layer MLP** [4], which is widely adopted in existing approaches. The “layered structure” mentioned by the reviewer is also a **common practice** in parameter-efficient fine-tuning [5,6]. Notably, some methods employ more than one adapter module to each ViT block or utilize more complicated adapter structures [7,8,9]. Section 3.3 (page 6, line 313) elaborates on the MoR Router, which also employs a **two-layer MLP** design. The noisy gating mechanism, described in Eq. 4, is a **standard technique** in MoE approaches [10].
>
> - We have validated the **individual contributions of the components with these simple and lightweight designs** in the ablation study in Table 3 (page 8) of the original submission. Here we supplement this by presenting the number of parameters and computational cost (measured by MACs, Multiply–accumulate operations) introduced by our modules, as shown in the table below. The calculations are based on a student model without decode heads, as heads are identical across configurations. The table indicates that **each TSAP branch accounts for less than 5% of the backbone’s parameters and computations** (4M *vs.* 86M parameters, 4G *vs.* 88G MACs). Similarly, **each MoR Router is as lightweight as 1.2M parameters and 1.2G MACs**.
>
>   The results confirm that **TSAP effectively preserves the representation biases from the teachers**, as evidenced by the higher similarity between the student and teachers (“Rep Sim”). Additionally, we validate that simply adding diverse representations (“N MoR”) does not lead to an overall improvement and may even fall behind the student without biases. With both components, SAK not only preserves the representation biases but also **maximizes multi-task performance across all five tasks**, with an overall improvement of 1.18%.
>
> | TSAP           | MoR         | #Param       | MACs         | Rep Sim    | Semseg         | Parsing        | Saliency       | Normal           | Boundary       | $\Delta_m\%$ |
> | -------------- | ----------- | ------------ | ------------ | ---------- | -------------- | -------------- | -------------- | ---------------- | -------------- | ------------ |
> |                |             | $\downarrow$ | $\downarrow$ | $\uparrow$ | mIoU$\uparrow$ | mIoU$\uparrow$ | maxF$\uparrow$ | mErr$\downarrow$ | odsF$\uparrow$ | $\uparrow$   |
> | N              | N           | 86M          | 88G          | 0.3344     | 80.97          | 69.71          | 84.64          | 14.11            | 72.82          | -1.21        |
> | Y (3 branches) | N           | 98M          | 100G         | 0.8708     | 81.26          | 69.92          | 84.31          | 14.45            | 71.41          | -2.03        |
> | Y (3 branches) | Y (5 tasks) | 104M         | 106G         | 0.8708     | **81.65**      | **72.38**      | **84.87**      | **14.05**        | **73.23**      | **-0.03**    |
>
> - Additionally, we would like to underscore that our framework does not rely on the specific structure of the TSAP and MoR Routers. It is compatible with various adapter and router designs, allowing for potential enhancements in model complexity and performance with more dedicated designs.

---

> ### Author Response · Authors · 2024-11-24
> **Response to Reviewer tiXZ (Part 2)**
>
> ### Weakness2
>
> We appreciate the reviewer’s detailed feedback regarding hyperparameter tuning. We would like to clarify that our experimental setups **basically follow prior work and do not involve hyperparameter tuning across different experiments**.
>
> - Specifically, for the first stage, we follow the setup of the state-of-the-art baseline RADIO for optimizer, learning rate, LR scheduler, and distillation loss (in Eq. 5, page 7, line 332), while adopting another baseline Theia’s batch size configuration since we use the same dataset. To enhance training efficiency, we reduce the number of epochs. For the second stage, we use the **identical hyperparameters** (including data augmentation and task-specific loss functions) as those in prior multi-task learning works [1,2,3]. We offered detailed specifications of implementation and experiments in Appendix B (page 21-23).
>
> - The only hyperparameter we have tried tuning is the balancing factor $\gamma$ in Eq. 6 (page 7, line 340), which balances the distillation loss and task losses during the second stage. Its impact was analyzed in Table 19 (Appendix C, page 26) and corresponding text (page 25, line 1347). For all other experiments, we use a default value of 1.0 for simplicity.
>
>
> In summary, we would like to underscore that **fixed hyperparameters are used across all experiments** (except Table 19 for experiments on varying $\gamma$​), confirming the contributions of our proposed methodology without extensive hyperparameter search.
>
>
>
> [1] Hanrong Ye and Dan Xu. Inverted pyramid multi-task transformer for dense scene understanding. In ECCV, 2022.
>
> [2] Hanrong Ye and Dan Xu. TaskExpert: Dynamically assembling multi-task representations with memorial mixture-of-experts. In ICCV, 2023a.
>
> [3] Yuqi Yang, Peng-Tao Jiang, Qibin Hou, Hao Zhang, Jinwei Chen, and Bo Li. Multi-task dense prediction via mixture of low-rank experts. In CVPR, 2024d.
>
> [4] Neil Houlsby, Andrei Giurgiu, Stanislaw Jastrzebski, Bruna Morrone, Quentin De Laroussilhe, Andrea Gesmundo, Mona Attariyan, and Sylvain Gelly. Parameter-efficient transfer learning for NLP. In ICML, 2019.
>
> [5] Minghao Fu, Ke Zhu, and Jianxin Wu. Dtl: Disentangled transfer learning for visual recognition. In AAAI, 2024.
>
> [6] Otniel-Bogdan Mercea, Alexey Gritsenko, Cordelia Schmid, and Anurag Arnab. Time-Memory-and Parameter-Efficient Visual Adaptation. In CVPR, 2024.
>
> [7] Zhe Chen, Yuchen Duan, Wenhai Wang, Junjun He, Tong Lu, Jifeng Dai, and Yu Qiao. Vision Transformer Adapter for Dense Predictions. In ICLR, 2023.
>
> [8] Dongshuo Yin, Xueting Han, Bin Li, Hao Feng, and Jing Bai. Parameter-efficient is not sufficient: Exploring parameter, memory, and time efficient adapter tuning for dense predictions. In ACM MM, 2024.
>
> [9] Fengze Jiang, Shuling Wang, and Xiaojin Gong. Task-Conditional Adapter for Multi-Task Dense Prediction. In ACM MM, 2024.
>
> [10] Noam Shazeer, Azalia Mirhoseini, Krzysztof Maziarz, Andy Davis, Quoc Le, Geoffrey Hinton, and Jeff Dean. Outrageously large neural networks: The sparsely-gated mixture-of-experts layer. In ICLR, 2017.

---

### Author Response · Authors · 2024-11-27
**General Response**

We sincerely appreciate all the reviewers for their time and effort in evaluating our submission and providing constructive feedback. We are greatly encouraged by the acknowledgement from all the reviewers with positive ratings.

We highly value the discussions with the reviewers and have integrated the new analysis and results during the discussion into our updated manuscript (Appendix F, marked as blue).

Please let us know if you have any additional comments; we would be happy to discuss them and incorporate your feedback into the revision. Once again, we appreciate all the reviewers for their valued feedback and helping improve our work.

---

### Meta-Review · Area_Chair_duym · 2024-12-15

**Metareview:**

This paper proposes a multi-task learning method to distill knowledge from a committee of Vision Foundation Models (VFMs). Unlike Unlike existing methods that use a single backbone for knowledge transfer, this paper preserves the unique representation bias of each teacher by collaborating the lightweight Teacher-Specific Adapter Path modules with the TeacherAgnostic Stem. It can effectively leverage the information from VFMs through a mixture-of-representations router. Experiments on multiple datasets have demonstrated the effectiveness of the method.

All reviewers are unanimously positive on this submission, and like the contributions including 1) the idea is somehow interesting as a multi-task learning framework to learn the knowledge from VFMs while preserving model biases; 2) the method is parameter and inference efficient; 3) the experiments are extensive and support the proposed claims; 4) the paper is well written. The common concerns from the reviewers are 1) architecture/complexity overhead; 2) missing experiments on other CV/VLM tasks; 3) missing a few more ablations/discussion and details. However, most of the concerns were well addressed after the rebuttal. The AC agrees with the reviewers' unanimous decision.

**Additional Comments On Reviewer Discussion:**

The common concerns from the reviewers are 1) architecture/complexity overhead; 2) missing experiments on other CV/VLM tasks; 3) missing a few more ablations/discussion and details. Most of the concerns were well addressed after the rebuttal. Two reviewers (gxTF and yJkf) didn't respond, but the authors have provided sufficient responses.

---

### Decision · Program_Chairs · 2025-01-22

Accept (Poster)